# PHOTON: SPEEDUP VOLUME UNDERSTANDING WITH EFFICIENT MULTIMODAL LARGE LANGUAGE MODELS

**Chengyu Fang**[1,2,*]   **Heng Guo**[2,3,*]   **Zheng Jiang**[1,2]   **Chunming He**[4]
**Xiu Li**[1,†]   **Minfeng Xu**[2,3,†]
[1]Tsinghua University   [2]DAMO Academy, Alibaba Group   [3]Hupan Lab   [4]Duke University

## ABSTRACT

Multimodal large language models are promising for clinical visual question answering tasks, but scaling to 3D imaging is hindered by high computational costs. Prior methods often rely on 2D slices or fixed-length token compression, disrupting volumetric continuity and obscuring subtle findings. We present Photon[1], a framework that represents 3D medical volumes with token sequences of variable length. Photon introduces instruction-conditioned token scheduling and surrogate gradient propagation to adaptively reduce tokens during both training and inference, which lowers computational cost while mitigating the attention dilution caused by redundant tokens. It incorporates a custom backpropagation rule with gradient restoration to enable differentiable optimization despite discrete token drop. To stabilize token compression and ensure reliable use of visual evidence, Photon further applies regularization objectives that mitigate language-only bias and improve reliability. Experiments on diverse medical visual question answering tasks show that Photon achieves state-of-the-art accuracy while reducing resource usage and accelerating both training and inference.

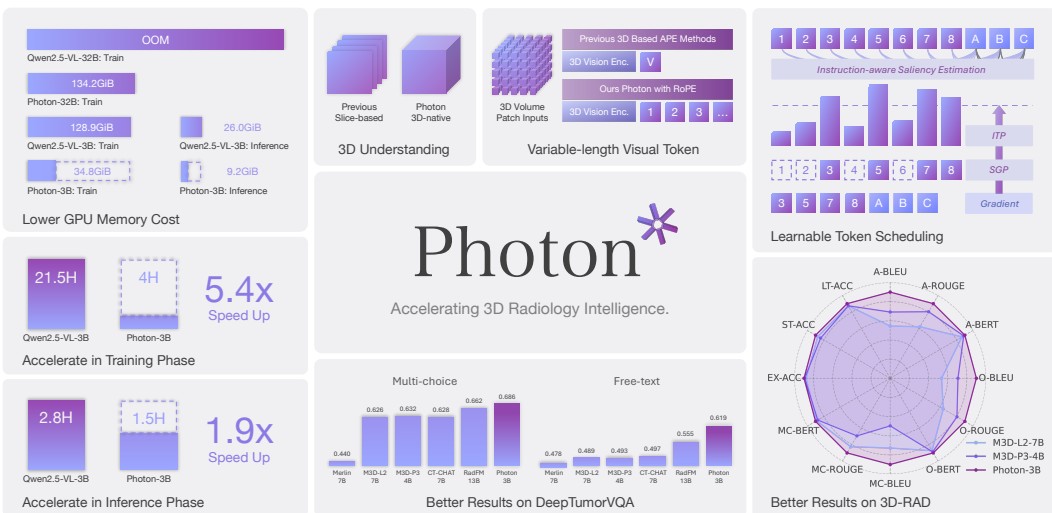

Figure 1: Photon is a 3D-native framework that adaptively models medical volumes using variable-length tokens, accelerating both training and inference. It enables efficient clinical question answering by removing instruction-irrelevant tokens while maintaining better quantitative performance.

---

*Equal Contribution, † Corresponding Author. This work was done during Chengyu Fang and Zheng Jiang's internship at DAMO Academy, Alibaba Group. Email: chengyufang.thu@gmail.com

[1]Photon: the quantum particle of light in nature and the fundamental component of X-rays in CT. Its speed, controllable energy deposition, and discrete interactions for precise measurement align well with our core idea.

# 1 INTRODUCTION

Recent advances in multimodal large language models (MLLMs) open new directions for clinical question answering. Despite these advances, modeling entire 3D volumes remains challenging because of heavy memory and computational demands. To reduce cost, many systems adopt slice-based pipelines or select a limited set of frames from CT or MRI scans (Xu et al., 2025; Chen et al., 2024a; Sellergren et al., 2025; Li et al., 2023a). Although such strategies shorten input sequences, they disrupt spatial continuity, eliminate volumetric detail, and introduce human bias in frame selection.

To preserve volumetric fidelity without manual frame selection, reducing visual tokens has become a promising strategy. In the general domain, approaches such as VisionZip (Yang et al., 2025d) and LLaVA-PruMerge (Shang et al., 2024) prune or merge tokens directly based on attention or similarity, achieving a speed-up in the inference time. However, these methods uniformly apply saliency signals without instruction awareness, making them prone to discarding subtle but clinically important patterns. Moreover, most employ fixed pruning ratios, limiting flexibility in instruction-conditioned tasks. ATP-LLaVA (Ye et al., 2025) introduces learnable thresholds for adaptive pruning, but still retains soft masks during training, so computation and memory are not reduced until inference.

In the medical domain, recent studies have advanced MLLMs toward volumetric modeling by aligning 3D vision encoders with large language models (Wu et al., 2023; Bai et al., 2024; Hamamci et al., 2024b; Blankemeier et al., 2024). These models support joint reasoning over full 3D volumes and alleviate some limitations of 2D slice-based inputs. However, they often produce visual embeddings in a few and fixed length, limiting high-resolution detail preservation and spatial interpretability.

To address these limitations, we propose Photon, a native 3D MLLM framework to model 3D medical volumes with variable-length token sequences, preserving volumetric fidelity without slice sampling or fixed-length token compression. Photon integrates two complementary components: Instruction-conditioned Token Scheduling (ITS) and Surrogate Gradient Propagation (SGP). ITS adaptively reduces visual tokens by estimating instruction-aware saliency and predicting instance-specific thresholds. SGP enables effective training by restoring gradients to retention probabilities, ensuring that token selection remains differentiable and instruction-driven. In the forward pass, Photon removes unimportant tokens together with their caches and positional encodings to lower computation. In the backward pass, a straight-through surrogate mask restores gradients to the underlying retention probabilities, allowing optimization to reflect instruction-driven token importance. To further enhance stability, Photon incorporates lightweight regularization objectives that mitigate language-only hallucination bias and improve reliability.

**The main contributions of this work are summarized as follows:**

(1) We present Photon, a medical multimodal large language model that directly operates on 3D scans and produces variable-length token sequences, preserving critical volumetric details without slice sampling or fixed token compression ratio.

(2) We introduce Instruction-conditioned Token Scheduling (ITS), which leverages instruction self-affinity and instruction-vision interactions to adaptively regulate visual token retention.

(3) We propose Surrogate Gradient Propagation (SGP), a mechanism that combines discrete token reduction in the forward pass with surrogate gradient optimization in the backward pass, enabling differentiable training despite discrete token dropping.

(4) Photon significantly accelerates both training and inference compared to general-purpose MLLMs, while achieving state-of-the-art performance on multiple medical visual question answering tasks.

# 2 RELATED WORK

**Medical Vision-Language Models.** Early medical VLMs, such as BiomedGPT (Luo et al., 2023) and CT2Rep (Hamamci et al., 2024a), primarily adopted CLIP-style alignment between 2D slices and text, but lacked scalability and capability of interactive reasoning. More recent MLLM models have also been adapted to the medical tasks, inspiring systems such as HealthGPT (Lin et al., 2025a), Med-R1 (Lai et al., 2025), and others (Li et al., 2023a; Sellergren et al., 2025; Pan et al., 2025). Most of these methods retain the slice-based paradigm, which undermines volumetric context and risks spatial discontinuity. To enhance spatial understanding, volumetric MLLMs (Lee et al., 2024) such

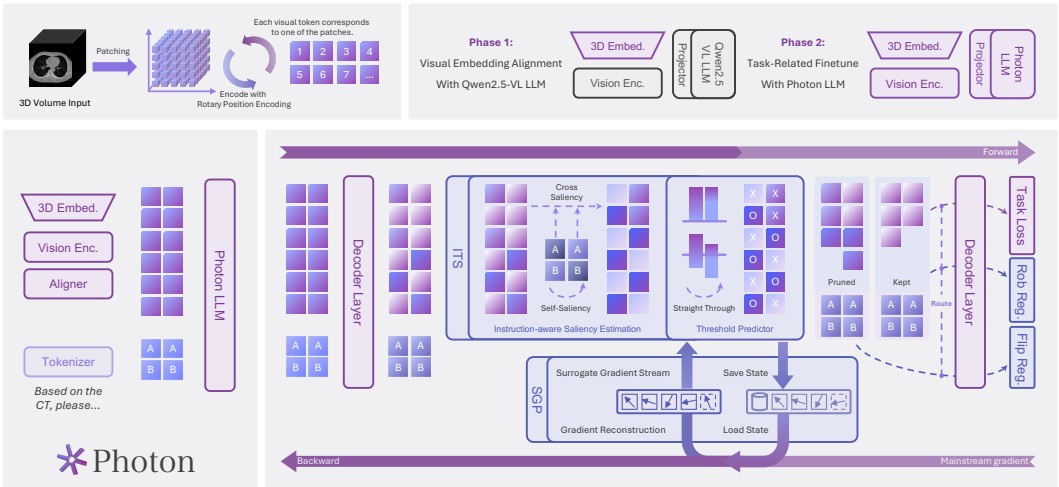

Figure 2: Photon's pipeline: Phase 1 aligns the visual embedding layer, and Phase 2 finetunes all modules for task adaptation, learning token reduction threshold estimation through our backpropagation strategy. Modules in right upper with black contour are not updated during training.

as RadFM (Wu et al., 2023) and M3D (Bai et al., 2024) align 3D vision encoders with language models, but compress each scan into a fixed-length sequence of vision tokens. The sequence length is typically small due to memory constraints, limiting their scalability to high-resolution inputs and weakening fine-grained spatial detail. OmniV-Med (Jiang et al., 2025) instead supports variable-length sequences, but its pruning strategy relies on L1 similarity of slice features, a coarse criterion that can inadvertently remove subtle, yet clinically important pathologies.

**Visual Token Reduction.** To lower computational cost in large vision-language models, many works compress the visual representation at the vision-language interface or within vision encoder layers. Representative designs include Q-Former (Li et al., 2023b), clustering-based merging (Shang et al., 2024), and pooling modules (Li et al., 2024b; Yao et al., 2024), as well as internal pruning that removes less informative tokens inside the model (Chen et al., 2024b). Although these approaches reduce token redundancy and improve efficiency, they ignore that different instructions require different amounts of information. In medical imaging, diverse instructions often focus on distinct anatomical regions represented by visual token sequences of varying lengths. Therefore, there is a pressing need for adaptive token reduction strategies tailored to computationally intensive scenarios. IVTP (Huang et al., 2024), TransPrune (Li et al., 2025a) and some pioneering works (Han et al., 2024; Wang et al., 2024b; Xing et al., 2024; Zhang et al., 2025a;c; Dhouib et al., 2025) sort the visual tokens by heuristic methods (Yang et al., 2025d; Alvar et al., 2025; Zhang et al., 2025d; Yang et al., 2025b; Sun et al., 2025) to enhance the relevance between visual token reduction and instructions. These methods account for instruction-vision correlation but are restricted to fixed retention ratios or manually set thresholds, which risks discarding vital pathological cues in medical imaging. ATP-LLaVA (Ye et al., 2025) introduces learnable thresholds to enable adaptive pruning but applies pruning only at inference, while additional computational cost is still required during training.

## 3 METHODOLOGY

Photon is a multimodal framework that integrates a 3D vision encoder with a large language model to jointly process volumetric scans and clinical instructions (see Fig. 2). It directly models entire medical volumes by partitioning them into 3D patches and mapping each patch to a token. The token sequence length is dynamically adjusted to balance spatial detail preservation against computational cost while mitigating the attention dilution caused by redundant tokens. **In the forward computation**, visual tokens are adaptively reduced according to their estimated importance and the predicted threshold derived from the visual summaries. **In the backward computation**, discrete selection remains differentiable through a straight-through path with masked gradient routing aligned to the active tokens, ensuring the same selection logic is honored in both training and inference.

## 3.1 INPUT PROCESSING AND 3D VISION ENCODER

Given an input scan of size $D \times H \times W$, the volume is divided into non-overlapping $14 \times 14 \times 14$ patches. Each patch is linearly embedded into a $d$-dimensional vector. To balance resolution and sequence length, spatial merging is applied only along $(H, W)$ with stride $S$, while the depth $D$ remains unchanged. The compact grid contains visual tokens with a length of:

$$N_v = D \cdot \frac{H'}{S} \cdot \frac{W'}{S}, \tag{1}$$

where $H'$ and $W'$ are the in-plane sizes after patchification. Let $\mathcal{V} = \{1, \ldots, N_v\}$ be the visual-token index set and $\mathcal{Q} = \{1, \ldots, N_t\}$ be the instruction-token index set. Our 3D vision encoder (ViT-based) then handles the full visual-token set $\mathcal{V}$, and encodes these tokens into visual embeddings $\text{T}_{\text{vis}}$ in the same hidden dimension of $\text{T}_{\text{txt}}$ that is encoded by a language tokenizer. Photon's LLM decoder part takes as input the following hybrid tokens:

$$\text{T}^0 = [\,\text{T}_{\text{vis}}^0, \, \text{T}_{\text{txt}}^0\,], \tag{2}$$

with positional indices following Qwen2.5-VL (Bai et al., 2025). Tokens are processed by self-attention and feed-forward blocks. To improve efficiency under instructions, Surrogate Gradient Propagation (SGP) is applied at selected layer $\ell \in \ell_n$ of Photon's language modules.

## 3.2 INSTRUCTION-CONDITIONED TOKEN SCHEDULING

Token-level saliency scores provide a relative ranking of visual tokens within each sample, but do not specify how many tokens should be retained in total. A fixed retention ratio would treat all scans and instructions equally, ignoring differences in complexity and clinical focus implied by instructions. To address this, we introduce Instruction-conditioned Token Scheduling (ITS), a mechanism that adapts the pruning threshold to each sample by conditioning on both the instruction and the token-level distribution. ITS integrates the Instruction-aware Saliency Estimation (ISE) rule with an Instance-aware Threshold Predictor (ITP), ensuring that token retention adapts to instruction demands while maintaining visual compactness.

**Instruction-aware Saliency Estimation.** To estimate the saliency of visual tokens under instruction guidance, we introduce an ISE at a selection layer $\ell \in \ell_n$, which explicitly links token saliency to variations in the given instructions.

Let $q_t$ denote the query vector of the $t$-th instruction token, $k_{t'}$ the key vector of the $t'$-th instruction token ($t, t' \in \mathcal{Q}$), and $\alpha$ the dimensions of the attention head. To prevent trivial self-matching, we exclude the diagonal ($t' = t$) and define the centrality score as:

$$c_t = \sum_{t' \in \mathcal{Q}} \max(\frac{\langle q_t, k_{t'} \rangle}{\sqrt{\alpha}}, 0), \tag{3}$$

which measures the extent to which token $t$ is positively reinforced by other instruction tokens, while ignoring negative contributions. The centralities are then normalized into weights:

$$w_t = \frac{c_t}{\sum_{t' \in \mathcal{Q}} c_{t'}}, \qquad \sum_{t \in \mathcal{Q}} w_t = 1, \tag{4}$$

where the tokens with consistently strong positive affinities obtain larger $w_t$, thereby highlighting salient instruction tokens that contribute more to the results. Let $k_j$ denote the key vector of the $j$-th ($j \in \mathcal{V}$) visual token, we compute the weighted instruction-visual saliency score by:

$$\rho_j = \text{norm}(u_j), \qquad u_j = \sum_{t \in \mathcal{Q}} w_t \frac{\langle q_t, k_j \rangle}{\sqrt{\alpha}}, \qquad \rho_j \in [0, 1], u_j \in [-\infty, +\infty] \tag{5}$$

where $\text{norm}(\cdot)$ denotes instance-wise min-max normalization. The saliency score $\rho_j$, reflecting how strongly each visual token aligns with the most central instruction tokens, provides a relative importance ranking that serves as the basis for adaptive thresholding.

**Instance-aware Thresholding Prediction.** While saliency scores provide token-wise rankings, they do not determine the retention level per instance. A fixed keep ratio ignores variability in scan complexity and instruction demands.

To adapt retention, Photon introduces ITP to predict an instance-specific threshold conditioned on both the instruction and the distribution of visual saliency scores. According to the results of ISE, $u = \{u_j\}_{j \in \mathcal{V}}$ is the raw logits and $\rho = \{\rho_j\}_{j \in \mathcal{V}}$ is the normalized scores. We construct a feature vector from three complementary views:

$$z = [\, \Psi(\rho), \ \Phi(u), \ \Upsilon(u) \,], \tag{6}$$

where $\Psi, \Phi, \Upsilon$ summarize distributional shape, absolute scale, and compressed tails, respectively (see Appendix A.1.2). We combine these to form a compact descriptor $z$. A lightweight predictor then maps $z$ to a scalar threshold:

$$\theta \;=\; \sigma\big(W_2\,\phi(W_1 z + b_1) + b_2\big), \qquad \theta \in [0, 1], \tag{7}$$

where $(W_1, b_1)$ and $(W_2, b_2)$ denote the weight-bias pairs of the hidden and output layers, $\phi$ is the GELU activation, and $\sigma$ is the logistic function. The resulting $\theta$ serves as a smooth probabilistic threshold, which can be directly compared with the normalized saliency $\rho_j$:

$$q_j \;=\; \sigma\left(\tfrac{\rho_j - \theta}{\tau_{ce}}\right), \qquad q_j \in [0, 1], \tag{8}$$

where the $q_j$ denotes the retention probability and $\tau_{ce}$ is a sufficiently small coefficient to make the continuous mask approximate to binary. Based on this sigmoid result, we split to get the token retention binary mask with:

$$M_j = \mathbf{1}\{\, q_j > 0.5 \,\}, \tag{9}$$

Tokens with $M_j = 0$ are hard dropped and removed entirely, including their cache, attention mask, and positional embedding. To enable differentiability, we replace $M$ with a straight-through surrogate:

$$\widetilde{M} = \mathrm{sg}(M) - \mathrm{sg}(q) + q, \tag{10}$$

where $\mathrm{sg}(\cdot)$ denotes the stop-gradient operator, and $q = \{q_j\}_{j \in \mathcal{V}}$. Thus the forward path follows hard selection, while the backward path propagates gradients through probabilities. The surrogate gradients that govern the update dynamics are detailed in the subsequent forward-backward analysis.

### 3.3 Surrogate Gradient Propagation

Instruction-conditioned Token Scheduling (ITS) produces the surrogate mask $\widetilde{M}$ that determines which tokens are retained in the forward sequence. Surrogate Gradient Propagation (SGP) addresses the training challenge that arises from this discrete selection by restoring gradients to the retained tokens and by providing surrogate gradients for the retention probabilities derived from the threshold predictor. Specifically, the upstream gradient of the compressed sequence is scattered back to the original tokens as:

$$\frac{\partial \mathcal{L}}{\partial \mathrm{T}_{vis}^{\ell}} = \mathcal{S}\left(\tfrac{\partial \mathcal{L}}{\partial \mathrm{T}_{vis}^{\ell}\,'}; \, \widetilde{M}\right), \tag{11}$$

where $\mathcal{S}(\cdot; \widetilde{M})$ scatters gradients from the compressed sequence $\mathrm{T}_{vis}^{\ell}\,'$ back to their original positions in $\mathrm{T}_{vis}^{\ell}$. This scattered path keeps decoder activations trainable for retained tokens. However, the threshold predictor receives no useful optimization signal, since the hard selection operation blocks gradients with respect to its probabilities. To address this, we design a surrogate gradient on $q_j$ so that the predictor can be optimized under token pruning.

With token-level gradients available, we define an importance score that measures the task-driven contribution of each token. Saliency scores $\rho_j$ and retention probabilities $q_j$ characterize relevance or selection tendency, but they do not reflect the effect of tokens on the loss. We therefore adopt a first-order approximation, where the variation in loss induced by removing a token is estimated by the inner product between its activation and gradient (detailed in Appendix A.1.4):

$$\eta_j = \Big\langle (\mathrm{T}_{vis}^{\ell})_j, \ \big(\tfrac{\partial \mathcal{L}}{\partial \mathrm{T}_{vis}^{\ell}}\big)_j \Big\rangle. \tag{12}$$

Here $\eta_j$ reflects the contribution of token $j$ to the loss, with larger values indicating stronger influence. These scores are standardized and clipped:

$$z_j = \frac{\eta_j - \mu(\eta \odot \widetilde{M})}{\max\{\mathrm{Std}(\eta \odot \widetilde{M}), \ \epsilon_{std}\}}, \qquad z_j \leftarrow \mathrm{clip}(z_j, -c, c), \tag{13}$$

where $\mu(\cdot)$ and $\text{Std}(\cdot)$ denote the masked mean and standard deviation, $\epsilon_{\text{std}}$ prevents division by small variance, and $c$ bounds outliers. The standardized values are rescaled into $(0, 1)$ through a smooth monotonic mapping $\psi(\cdot)$. The directional term is then defined as:

$$r_j = \psi(z_j), \qquad d_j = 0.5 - \text{sg}(r_j), \tag{14}$$

where $\psi(0) = 0.5$ serves as the neutral point, so that $d_j$ quantifies the deviation from this midpoint. In parallel, a magnitude factor is computed by accumulating activation-gradient products across the feature dimension:

$$m_j = \sum_{k=1}^{d} \left|(\text{T}^\ell_{\text{vis}})_{jk}\, g_{jk}\right|, \qquad g_j = \left(\frac{\partial \mathcal{L}}{\partial \text{T}^\ell_{\text{vis}}}\right)_j, \tag{15}$$

where $(\text{T}^\ell_{\text{vis}})_{jk}$ and $g_{jk}$ are the $k$-th components of $(\text{T}^\ell_{\text{vis}})_j$ and $g_j$. These values are normalized within each row and clamped to a bounded interval:

$$s_j = \text{clip}\left(\frac{m_j}{\mu_{\text{row}}(m \odot \widetilde{M}) + \varepsilon},\, s_{\text{min}},\, s_{\text{max}}\right), \tag{16}$$

which yields a scale factor that amplifies consistently strong activation-gradient interactions while suppressing weak ones. Finally, combining direction and magnitude provides the surrogate gradient with respect to the relaxed probabilities:

$$\frac{\partial \mathcal{L}}{\partial q_j} \approx \beta\, d_j\, s_j\, \max\{q_j(1 - q_j), \epsilon_{sat}\}. \tag{17}$$

where $\beta$ balances directional and magnitude contributions, $\epsilon_{sat} > 0$ provides a lower bound on the gradient term $q_j(1 - q_j)$ to avoid vanishing when $q_j$ is close to 0 or 1, thereby ensuring stable updates. As a result, tokens assessed as more informative are gradually reinforced toward retention, whereas less informative tokens are progressively suppressed, which guarantees that the pruning mechanism behaves as intended (see detailed analysis in Appendix A.1.3).

## 3.4 OBJECTIVE

**Soft Retention Band.** While forward-backward learning already propagates supervision to guide pruning, unconstrained training may converge to degenerate cases of keeping nearly all or pruning excessively. To avoid such extremes, we constraint the retention probabilities $q_j$ by softly constraining their mean ratio $r$ within a band:

$$\mathcal{L}_{\text{band}} = \mathbb{E}[\max(0, r - r_{\text{max}}) + \max(0, r_{\text{min}} - r)], \qquad r = \frac{1}{N_v}\sum_{j \in \mathcal{V}} q_j, \tag{18}$$

This band-constrained penalty softly restricts the retention ratio to a stable range, ensuring pruning remains effective without overriding the dynamics of the forward-backward process.

**Robustness Regularizer.** Another failure mode in multimodal learning is language-only hallucination bias, where answers rely on textual priors while ignoring the image. This risk is critical in medical tasks, since predictions may appear correct without visual grounding. To address it, the model is encouraged to produce higher predictive uncertainty whenever visual evidence is insufficient. Given a perturbed volume $\tilde{x}$ (by masking, shuffling, etc.), the robustness loss is defined as:

$$\mathcal{L}_{\text{robust}} = -\mathbb{E}_{\tilde{x}}[H(p_\theta(\cdot|\tilde{x}))], \qquad H(p) = -\sum_v p(v)\log p(v). \tag{19}$$

During these perturbed cases, we drop other loss functions and optimize only $\mathcal{L}_{\text{robust}}$. This discourages overconfident predictions without valid visual grounding.

**Flip Regularizer.** To further reduce reliance on spurious pruning patterns, we introduce a mask-flip regularizer. With a certain probability, the retention mask is inverted so that tokens originally reduced are retained and those originally retained are reduced. Under this adversarial mask, the model should not remain highly confident on the correct answer; otherwise, it indicates over-reliance on textual priors or shortcut cues. We penalize such cases by assigning a higher loss whenever the model still predicts the correct label with high confidence:

$$\mathcal{L}_{\text{flip}} = -\mathbb{E}_{\widetilde{M}^{\text{flip}}}\left[\log\left(1 - p_\theta(y \mid x, \widetilde{M}^{\text{flip}}) + \epsilon\right)\right], \tag{20}$$

Table 1: Comparison of zero-shot and finetuned performance across six tasks in 3D-RAD benchmark. **violet** and **indigo** indicate the best and the second best. Stat. Temp. Diag. = Static Temporal Diagnosis, Longit. Temp. Diag. = Longitudinal Temporal Diagnosis.

| Task | Metric | Zero-shot | | | | | | Finetuned | | |
|------|--------|-----------|---|---|---|---|---|-----------|---|---|
| | | Qwen2.5-VL 3B | RadFM 13B | M3D-L2 7B | M3D-P3 4B | OmniV 1.5B | Lingshu 7B | M3D-L2 7B | M3D-P3 4B | Photon 3B |
| Existence Detection | Accuracy | 19.52 | 29.20 | 18.00 | 40.25 | 28.66 | 59.60 | 81.09 | 82.43 | 83.07 |
| Stat. Temp. Diag. | Accuracy | 0.00 | 44.11 | 25.47 | 25.40 | 22.96 | 6.02 | 51.20 | 49.30 | 52.86 |
| Longit. Temp. Diag. | Accuracy | 0.29 | 42.99 | 24.17 | 24.31 | 24.23 | 12.13 | 74.78 | 74.77 | 77.01 |
| Medical Measurement | BLEU | 1.78 | 3.34 | 15.95 | 2.55 | 2.52 | 2.50 | 30.54 | 33.52 | 37.74 |
| | ROUGE | 4.30 | 6.62 | 23.24 | 5.63 | 7.88 | 5.45 | 36.06 | 36.46 | 39.36 |
| | BERT Score | 84.20 | 86.85 | 91.50 | 85.74 | 85.66 | 83.81 | 94.65 | 94.86 | 96.14 |
| Image Observation | BLEU | 3.51 | 13.48 | 10.69 | 16.31 | 16.42 | 5.34 | 31.28 | 39.66 | 51.59 |
| | ROUGE | 8.84 | 19.14 | 20.82 | 23.19 | 26.69 | 12.25 | 39.12 | 50.52 | 56.66 |
| | BERT Score | 84.53 | 87.16 | 86.61 | 86.92 | 88.29 | 85.67 | 90.00 | 92.19 | 93.62 |
| Anomaly Detection | BLEU | 2.93 | 11.00 | 9.10 | 15.06 | 13.47 | 3.71 | 25.25 | 33.28 | 42.33 |
| | ROUGE | 9.17 | 17.62 | 18.64 | 23.19 | 25.72 | 9.46 | 33.76 | 42.45 | 47.50 |
| | BERT Score | 84.47 | 86.76 | 86.07 | 87.11 | 88.21 | 84.81 | 89.16 | 90.72 | 91.96 |

where $p_\theta(y \mid x, \widetilde{M}^{\mathrm{flip}})$ is the probability of the ground-truth label under the flipped mask and $\epsilon$ is a small constant for numerical stability. This regularizer discourages the model from producing correct answers under inconsistent retention, thereby reinforcing reliance on the intended pruning structure.

**Overall Objective.** In training Phase 1, we only use the cross-entropy loss to supervise the alignment. In training Phase 2, the complete training objective combines faithful language modeling, stable token usage, robustness against language-only hallucination bias, and consistency under mask flipping:

$$\mathcal{L} = (\mathcal{L}_{\mathrm{CE}} \text{ or } \mathcal{L}_{\mathrm{band}} \text{ or } \mathcal{L}_{\mathrm{robust}}) + \mathcal{L}_{\mathrm{flip}}, \qquad \mathcal{L}_{\mathrm{CE}} = -\mathbb{E}_{(x,y)}[\log p_\theta(y|x)]. \tag{21}$$

This innovative design ensures that Photon learns to generate clinically accurate text, maintains efficiency through adaptive token pruning, and grounds its reasoning on volumetric evidence rather than spurious textual patterns.

## 4 EXPERIMENTS

All comparison baselines are initialized from models that have been pretrained or finetuned on large-scale medical datasets and then further finetuned following their respective protocols. In contrast, before task-specific finetuning, our method applies only a lightweight alignment of the 3D vision patch embedding layer using volume-caption pairs while keeping the ViT backbone, MLP aligner, and LLM decoder frozen. Details of the training process (Appendix A.1.1), experimental setup (Appendix A.2.1), and dataset information (Appendix A.2.2) are provided in the appendix.

### 4.1 COMPARATIVE EVALUATION

On 3D-RAD (Gai et al., 2025), Photon achieves the best finetuned results across all tasks in Table 1. The most notable gains appear in anomaly detection and image observation, where performance improves by about 14.0% compared with the best baseline. Medical measurement also shows a clear improvement of about 7.3%, which is critical since measurement accuracy directly underpins many downstream diagnostic decisions. Longitudinal temporal diagnosis rises by around 3.0%. These results highlight Photon's ability to retain volumetric details and adjust token selection based on instruction relevance, thereby enhancing the accuracy of both descriptive and quantitative clinical predictions. The consistent improvements across descriptive (e.g., anomaly detection, observation), quantitative (e.g., measurement), and reasoning-oriented (e.g., temporal diagnosis) tasks indicate that Photon generalizes beyond a single evaluation type.

On DeepTumorVQA (Chen et al., 2025c), Photon also obtains the highest overall accuracy in both multi-choice and free-text settings in Table 2. The total average in multi-choice increases by about 3.6%, while in free-text the improvement is larger at approximately 11.5%. The strongest gains occur in measurement subtypes evaluated with MRA, where Photon improves by more than 35.3%, confirming its strength in handling numerical and quantitative queries. Visual reasoning subtypes,

Table 2: Performance on DeepTumorVQA benchmark. Subtypes marked with * indicate free-text numerical answers evaluated using MRA, higher is better. Meas. = Measurement, Recog. = Recognition, Vis. Rsn. = Visual Reasoning, Med. Rsn. = Medical Reasoning.

| Type | Subtype | Multi-choice | | | | | | | Free-text | | | | | |
| | | Rand | Merlin 7B | M3D-L2 7B | M3D-P3 4B | CT-CHAT 7B | RadFM 13B | Photon 3B | Merlin 7B | M3D-L2 7B | M3D-P3 4B | CT-CHAT 7B | RadFM 13B | Photon 3B |
| | | — | | | | | | | | | | | | |
| Meas. | lesion volume measurement* | 0.250 | 0.253 | 0.815 | 0.825 | 0.833 | 0.815 | 0.825 | 0.079 | 0.085 | 0.079 | 0.075 | 0.112 | 0.265 |
| | organ HU measurement* | 0.250 | 0.254 | 0.638 | 0.640 | 0.637 | 0.647 | 0.642 | 0.487 | 0.490 | 0.491 | 0.513 | 0.608 | 0.777 |
| | organ volume measurement* | 0.250 | 0.262 | 0.747 | 0.754 | 0.750 | 0.755 | 0.756 | 0.526 | 0.535 | 0.528 | 0.549 | 0.583 | 0.717 |
| | Average | 0.250 | 0.256 | 0.733 | 0.740 | 0.740 | 0.739 | 0.741 | 0.364 | 0.370 | 0.366 | 0.379 | 0.434 | 0.587 |
| Recog. | colon lesion existence | 0.500 | 0.859 | 0.859 | 0.859 | 0.859 | 0.856 | 0.858 | 0.859 | 0.859 | 0.859 | 0.859 | 0.893 | 0.881 |
| | kidney cyst existence | 0.500 | 0.797 | 0.797 | 0.797 | 0.797 | 0.861 | 0.910 | 0.797 | 0.797 | 0.797 | 0.797 | 0.864 | 0.917 |
| | kidney lesion existence | 0.500 | 0.495 | 0.510 | 0.501 | 0.514 | 0.668 | 0.717 | 0.511 | 0.515 | 0.490 | 0.507 | 0.692 | 0.624 |
| | kidney tumor existence | 0.500 | 0.564 | 0.574 | 0.574 | 0.574 | 0.886 | 0.896 | 0.574 | 0.574 | 0.574 | 0.574 | 0.890 | 0.895 |
| | liver lesion existence | 0.500 | 0.535 | 0.524 | 0.517 | 0.524 | 0.652 | 0.681 | 0.524 | 0.524 | 0.524 | 0.524 | 0.662 | 0.640 |
| | pancreatic lesion existence | 0.500 | 0.718 | 0.718 | 0.718 | 0.718 | 0.810 | 0.805 | 0.718 | 0.718 | 0.718 | 0.718 | 0.871 | 0.819 |
| | Average | 0.500 | 0.661 | 0.664 | 0.661 | 0.664 | 0.789 | 0.811 | 0.664 | 0.665 | 0.660 | 0.663 | 0.812 | 0.796 |
| Vis. Rsn. | adjacent organ | 0.333 | 0.217 | 0.565 | 0.609 | 0.609 | 0.609 | 0.696 | 0.174 | 0.174 | 0.304 | 0.304 | 0.435 | 0.304 |
| | inter-segment comparison | 0.333 | 0.470 | 0.567 | 0.576 | 0.572 | 0.591 | 0.549 | 0.577 | 0.561 | 0.592 | 0.589 | 0.456 | 0.359 |
| | kidney volume comparison | 0.333 | 0.347 | 0.370 | 0.364 | 0.372 | 0.386 | 0.370 | 0.350 | 0.370 | 0.356 | 0.370 | 0.386 | 0.396 |
| | largest lesion attenuation | 0.333 | 0.317 | 0.541 | 0.539 | 0.544 | 0.555 | 0.539 | 0.526 | 0.544 | 0.548 | 0.542 | 0.521 | 0.571 |
| | largest lesion diameter* | 0.250 | 0.263 | 0.778 | 0.783 | 0.781 | 0.766 | 0.776 | 0.182 | 0.209 | 0.233 | 0.269 | 0.232 | 0.310 |
| | largest lesion location | 0.392 | 0.307 | 0.310 | 0.310 | 0.340 | 0.340 | 0.356 | 0.359 | 0.353 | 0.337 | 0.353 | 0.334 | 0.350 |
| | largest lesion slice* | 0.250 | 0.241 | 0.672 | 0.684 | 0.672 | 0.664 | 0.703 | 0.524 | 0.533 | 0.510 | 0.513 | 0.672 | 0.723 |
| | lesion count by location* | 0.250 | 0.583 | 0.861 | 0.860 | 0.862 | 0.861 | 0.865 | 0.534 | 0.534 | 0.534 | 0.534 | 0.506 | 0.546 |
| | lesion counting* | 0.328 | 0.455 | 0.781 | 0.784 | 0.796 | 0.790 | 0.985 | 0.000 | 0.000 | 0.000 | 0.000 | 0.001 | 0.917 |
| | lesion outlier | 0.500 | 0.521 | 0.507 | 0.549 | 0.451 | 0.493 | 0.535 | 0.451 | 0.535 | 0.535 | 0.577 | 0.521 | 0.634 |
| | liver lesion clustering | 0.333 | 0.331 | 0.438 | 0.475 | 0.463 | 0.469 | 0.475 | 0.388 | 0.469 | 0.469 | 0.431 | 0.513 | 0.456 |
| | organ aggregation* | 0.250 | 0.257 | 0.660 | 0.667 | 0.655 | 0.661 | 0.660 | 0.577 | 0.569 | 0.586 | 0.574 | 0.621 | 0.798 |
| | organ enlargement | 0.500 | 0.736 | 0.736 | 0.736 | 0.736 | 0.746 | 0.791 | 0.736 | 0.736 | 0.736 | 0.736 | 0.759 | 0.845 |
| | tumor organ HU difference* | 0.305 | 0.296 | 0.836 | 0.839 | 0.821 | 0.821 | 0.829 | 0.113 | 0.122 | 0.139 | 0.197 | 0.189 | 0.207 |
| | Average | 0.335 | 0.382 | 0.616 | 0.627 | 0.620 | 0.625 | 0.653 | 0.392 | 0.408 | 0.420 | 0.428 | 0.439 | 0.530 |
| Med. Rsn. | fatty liver | 0.333 | 0.318 | 0.461 | 0.455 | 0.481 | 0.481 | 0.558 | 0.481 | 0.481 | 0.396 | 0.487 | 0.578 | 0.773 |
| | lesion type classification | 0.500 | 0.865 | 0.865 | 0.865 | 0.865 | 0.865 | 0.865 | 0.865 | 0.865 | 0.865 | 0.865 | 0.851 | 0.865 |
| | pancreatic cyst resectability | 0.500 | 0.371 | 0.657 | 0.800 | 0.800 | 0.771 | 0.771 | 0.800 | 0.800 | 0.800 | 0.800 | 0.771 | 0.743 |
| | pancreatic lesion resectability | 0.333 | 0.379 | 0.483 | 0.483 | 0.483 | 0.483 | 0.483 | 0.414 | 0.483 | 0.483 | 0.483 | 0.483 | 0.483 |
| | pancreatic steatosis | 0.500 | 0.526 | 0.526 | 0.513 | 0.513 | 0.579 | 0.526 | 0.526 | 0.526 | 0.526 | 0.526 | 0.658 | 0.671 |
| | pancreatic tumor staging | 0.250 | 0.216 | 0.351 | 0.243 | 0.189 | 0.324 | 0.460 | 0.216 | 0.216 | 0.297 | 0.135 | 0.432 | 0.460 |
| | Average | 0.403 | 0.446 | 0.557 | 0.560 | 0.555 | 0.584 | 0.611 | 0.550 | 0.562 | 0.561 | 0.549 | 0.629 | 0.666 |
| | **Total Average** | 0.369 | 0.440 | 0.626 | 0.632 | 0.628 | 0.662 | 0.686 | 0.478 | 0.489 | 0.493 | 0.497 | 0.555 | 0.619 |

such as lesion counting and slice localization, also show large margins exceeding 20.7%, reflecting better spatial understanding. These large improvements in both measurement and reasoning tasks suggest that Photon is not only effective for recognition but also excels in tasks that require quantitative precision and spatial analysis. Taken together, the results from both 3D-RAD and DeepTumorVQA confirm that Photon delivers consistent and meaningful benefits in multiple tasks, strengthening its reliability as a versatile medical vision-language system.

### 4.1.1 COMPARATIVE EVALUATION WITH TOKEN PRUNING METHODS

Most existing pruning methods for multimodal large language models do not provide case-specific adaptive retention based on instruction-related visual content. Instead, they typically rely on fixed retention ratios. To fairly compare with such methods, we follow their configurations and evaluate them at three nominal retention levels (about 30%, 50%, and 70% of visual tokens), which correspond to keeping roughly 2.1K, 3.5K, and 4.9K vision tokens per sample in our 3D-RAD setup.

It is also important to note that these methods were not designed to optimize training-time acceleration, backpropagation stability, or overall training robustness. Several of them do not provide a complete training recipe. In the 3D medical setting, where the base model has limited prior knowledge and relies heavily on visual evidence, directly training with such pruning strategies can lead to premature removal of important tokens and a collapse towards language-only bias. To avoid this issue and keep the comparison fair, we first train a full Qwen2.5-VL model without pruning using the paradigms recommended in the respective papers, and then apply the pruning methods only at inference time using their official code. This isolates the effect of pruning at prediction time and avoids confounding factors from unstable training.

Most previous pruning methods, including VisionZip (Yang et al., 2025d) and HiPrune (Liu et al., 2025b), are not optimized for compatibility with FlashAttention in the visual encoder or require extra attention computation across all visual encoder layers. This leads to additional overhead and erodes the expected speed gains. In our setting, which has a large number of visual tokens and relatively short textual answers, these methods do not provide clear advantages in inference speed, HiPrune (Liu et al., 2025b) in particular slows down decoding despite keeping fewer tokens. By contrast, Photon couples instruction-conditioned pruning with a FlashAttention-friendly implementation and a pruning

Table 3: Comparison of different token pruning strategies on 3D-RAD. E.D. = Existence Detection, S.T.D. = Static Temporal Diagnosis, L.T.D. = Longitudinal Temporal Diagnosis.

| Methods | E.D. Acc. | S.T.D. Acc. | L.T.D. Acc. | Medical Measurement | | | Image Observation | | | Anomaly Detection | | | Infer Speed (Tok/s) | Token Num (Tok/case) |
|---|---|---|---|---|---|---|---|---|---|---|---|---|---|---|
| | | | | BLEU | ROUGE | BERT | BLEU | ROUGE | BERT | BLEU | ROUGE | BERT | | |
| Qwen2.5-VL | 81.97 | 47.62 | 75.36 | 37.00 | 38.57 | 96.04 | 52.72 | 56.48 | 93.63 | 42.01 | 47.36 | 91.93 | 2.30 | 7.0K |
| VisionZip | 82.00 | 47.19 | 75.42 | 37.04 | 38.69 | 96.06 | 52.51 | 56.36 | 93.63 | 41.96 | 47.39 | 91.93 | 2.32 | 2.1K |
| VisionZip | 81.99 | 47.74 | 75.93 | 37.11 | 38.68 | 96.07 | 52.72 | 56.60 | 93.65 | 41.98 | 47.36 | 91.92 | 2.23 | 3.5K |
| VisionZip | 82.01 | 47.40 | 75.88 | 37.07 | 38.82 | 96.04 | 52.57 | 56.43 | 93.64 | 42.01 | 47.39 | 91.93 | 2.18 | 4.9K |
| HiPrune | 81.99 | 48.08 | 75.50 | 37.00 | 38.59 | 96.05 | 52.56 | 56.46 | 93.64 | 42.08 | 47.32 | 91.93 | 0.76 | 2.1K |
| HiPrune | 81.97 | 47.84 | 75.58 | 37.14 | 38.75 | 96.07 | 52.65 | 56.60 | 93.63 | 41.92 | 47.31 | 91.91 | 0.75 | 3.5K |
| HiPrune | 82.02 | 47.19 | 75.35 | 37.00 | 38.62 | 96.06 | 52.57 | 56.44 | 93.62 | 41.94 | 47.40 | 91.91 | 0.73 | 4.9K |
| Photon | 83.07 | 52.86 | 77.01 | 37.74 | 39.36 | 96.14 | 51.59 | 56.66 | 93.62 | 42.33 | 47.50 | 91.96 | 4.12 | Dynamic |
| Photon MAX | 85.50 | 57.79 | 79.70 | 39.44 | 40.96 | 96.17 | 52.83 | 58.72 | 93.70 | 42.85 | 48.89 | 92.00 | 2.60 | Dynamic |

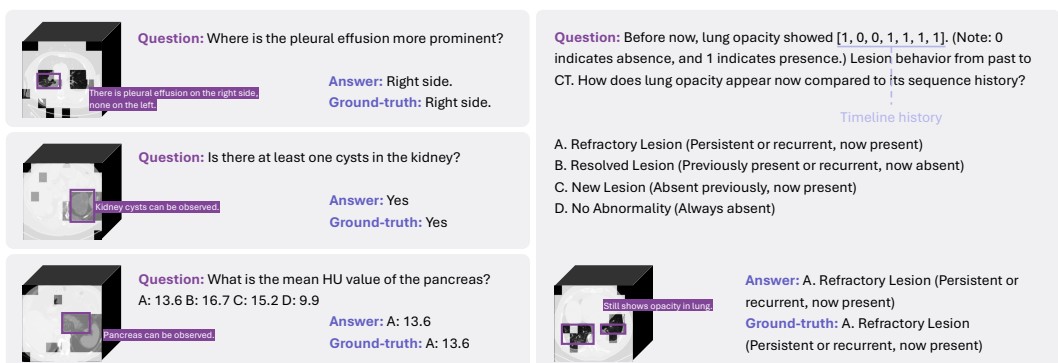

Figure 3: Visualization of Photon's results. White regions indicate reduced tokens, while purple boxes highlight retained areas that carry clinically relevant information for answering the questions.

schedule that is used during training. As shown in Table 3, Photon achieves higher accuracy on all three 3D-RAD tasks and reaches up to 4.12 tokens/s, improving both performance and inference efficiency under the same backbone.

## 4.2 VISUALIZATION ANALYSIS

Fig. 3 and Fig. 7 show qualitative results of Photon and its token reduction behavior. The examples demonstrate that the model is able to discard large portions of non-essential context while preserving subtle but diagnostically important structures. For instance, when asked about pleural effusion, Photon selectively retains regions around the thoracic cavity, while in kidney cystic-related questions, it keeps the cystic kidney area. These cases indicate that pruning decisions adapt to the clinical focus of each question rather than being uniform. The visual results combined with the results from Table 4 show that our token scheduling process reinforces the contribution of retained tokens, ensuring that relevant regions dominate the final prediction. As a result, Photon maintains high accuracy in various medical visual question answering tasks while reducing redundancy in visual tokens.

## 4.3 ABLATION STUDY

As shown in Table 4, Photon Phase 1, which finetunes only the modified 3D visual embedding layer, yields clear gains over the zero-shot initialization of Qwen2.5-VL (Bai et al., 2025): BLEU improves by 70% on anomaly detection, 134% on image observation, and existence detection accuracy rises by 44%. These results indicate that re-aligning the adapted 3D patch embedding provides a stronger initialization for subsequent finetuning. To test whether adapting the full vision stack is advantageous, we continued training from the Phase 1 checkpoint. However, even with a small learning rate, fully finetuning the vision encoder and MLP aligner (Vis. Ful. Ft.) led to overfitting and degraded performance. As illustrated in Fig. 4, the model also lost instruction-following capability and often produced answers inconsistent with the given questions.

Table 4: Ablation study on 3D-RAD. Computational statistics include training speed (iterations/s), average retaining tokens, and inference speed (tokens/s), inference peak GPU memory. E.D. = Existence Detection, S.T.D. = Static Temporal Diagnosis, L.T.D. = Longitudinal Temporal Diagnosis.

| Task | E.D. Acc. | S.T.D. Acc. | L.T.D. Acc. | Medical Measurement | | | Image Observation | | | Anomaly Detection | | | Computational Metrics | | | |
|---|---|---|---|---|---|---|---|---|---|---|---|---|---|---|---|---|
| | | | | BLEU | ROUGE | BERT | BLEU | ROUGE | BERT | BLEU | ROUGE | BERT | Train spd. | Train Tok. | Infer Spd. | Infer Mem. |
| *Without task-related finetune.* | | | | | | | | | | | | | | | | |
| Qwen2.5-VL | 19.52 | 0.00 | 0.29 | 1.78 | 4.30 | 84.20 | 3.51 | 8.84 | 84.53 | 2.93 | 9.17 | 84.47 | — | — | — | — |
| Vis. Ful. Ft. | 0.00 | 0.00 | 0.00 | 5.02 | 9.91 | 84.82 | 1.02 | 2.96 | 81.46 | 0.83 | 2.61 | 80.83 | — | — | — | — |
| Photon Phase 1 | 28.10 | 8.41 | 10.90 | 4.81 | 9.69 | 85.35 | 5.98 | 17.62 | 86.39 | 6.84 | 15.64 | 85.98 | — | — | — | — |
| *Finetuned with 3D-RAD datasets.* | | | | | | | | | | | | | | | | |
| Qwen2.5-VL Ft. | 81.97 | 47.62 | 75.36 | 37.00 | 38.57 | 96.04 | 52.72 | 56.48 | 93.63 | 42.01 | 47.36 | 91.93 | 0.15 | 7.00K | 2.30 | 26.0GiB |
| w/o. Photon Phase 1 | 82.35 | 52.20 | 75.69 | 39.10 | 40.53 | 96.17 | 51.23 | 56.27 | 93.55 | 42.13 | 47.46 | 91.92 | 0.84 | 0.45K | 4.09 | 9.2GiB |
| $\ell = 0$ | 82.09 | 47.21 | 75.45 | 40.29 | 41.69 | 96.34 | 52.18 | 57.16 | 93.66 | 42.12 | 47.03 | 91.89 | 0.82 | 0.74K | 4.03 | 9.2GiB |
| $\ell = \ell_{n/2}$ | 82.23 | 50.98 | 76.62 | 38.08 | 39.43 | 96.14 | 51.94 | 56.43 | 93.68 | 42.05 | 47.26 | 91.89 | 0.84 | 0.47K | 4.09 | 9.2GiB |
| $\ell = \ell_{3n/4}$ | 82.97 | 52.71 | 75.86 | 37.01 | 38.51 | 96.06 | 52.11 | 56.76 | 93.64 | 41.90 | 47.03 | 91.87 | 0.83 | 0.39K | 4.06 | 9.2GiB |
| w/o. ITS & SGP | 82.40 | 49.60 | 75.50 | 37.60 | 39.10 | 96.13 | 51.60 | 56.40 | 93.65 | 42.10 | 47.40 | 91.90 | 0.64 | 1.00K | 4.03 | 9.2GiB |
| w/o. Self-affinity | 82.34 | 52.82 | 77.89 | 37.41 | 38.78 | 96.05 | 51.78 | 55.99 | 93.67 | 41.97 | 47.46 | 91.91 | 0.82 | 0.54K | 4.08 | 9.2GiB |
| w/o. Robust Reg. | 81.81 | 48.18 | 77.38 | 37.54 | 39.25 | 95.97 | 51.70 | 56.46 | 93.59 | 42.05 | 46.90 | 91.95 | 0.87 | 0.29K | 4.14 | 9.2GiB |
| w/o. Flip Reg. | 82.20 | 52.09 | 77.09 | 36.97 | 38.24 | 95.96 | 51.79 | 56.61 | 93.64 | 42.02 | 47.50 | 91.92 | 0.85 | 0.38K | 4.12 | 9.2GiB |
| Photon | 83.07 | 52.86 | 77.01 | 37.74 | 39.36 | 96.14 | 51.59 | 56.66 | 93.62 | 42.33 | 47.50 | 91.96 | 0.85 | 0.39K | 4.12 | 9.2GiB |
| Photon Max | 85.50 | 57.79 | 79.70 | 39.44 | 40.96 | 96.17 | 52.83 | 58.72 | 93.70 | 42.85 | 48.89 | 92.00 | 0.19 | 3.08K | 2.60 | 21.4GiB |

**Question:** What device is observed on the chest wall?

**Ground-truth:** Pacemaker.

🙂 **Qwen2.5-VL:** A stethoscope is observed on the chest wall in the image.

😖 **Vis. Ful. Ft.:** A 24 mm diameter metallic stent is observed in the trachea.

🙂 **Photon Phase 1:** A device is observed on the chest wall, which appears to be a pacemaker.

**Question:** Which organ shows steatosis?

**Ground-truth:** Liver.

🙂 **Qwen2.5-VL:** The liver shows steatosis, which is characterized by the presence of ...

😖 **Vis. Ful. Ft.:** Trachea and both main bronchi are open. No occlusive pathology was ...

🙂 **Photon Phase 1:** The liver shows steatosis.

Figure 4: Visualization Results of base model and different visual alignment methods. Our method achieves alignment while avoiding mode collapse. Vis. Ful. Ft. = Visual Modules Fully Finetuned.

In the task-related finetuning stage, compared with other chosen layers, Photon achieves the best trade-off by placing the module at the $\ell = \ell_{n/4}$, attaining higher accuracy while maintaining efficiency. The importance of the core components is evident, removing ITS and SGP lowers static temporal diagnosis accuracy by more than 3%, increases the number of retained tokens, and slows training, confirming their necessity for both performance and efficiency. Regularization is also critical, since removing these terms not only reduces accuracy but also destabilizes training and undermines result reliability (Appendix A.2.3). Overall, Photon improves accuracy while substantially reducing computation. Compared with finetuned Qwen2.5-VL (Bai et al., 2025), it consistently achieves better performance while reducing inference memory usage by about two-thirds and achieving over a five-fold speedup. Gains in inference speed are more modest due to the use of KV Cache, but remain steady.

Beyond the main Photon design, we offer an alternative option called Photon Max. It reuses the original visual patch embedding layer of Qwen2.5-VL (Bai et al., 2025) with our components, improving existence detection and static temporal diagnosis by 2.4% and 4.9%, respectively, while training about 25% faster than Qwen2.5-VL with lower memory usage, thus serving as a more accuracy-oriented variant that remains more efficient than the original baseline.

## 5 CONCLUSION

We present Photon, an efficient MLLM framework for 3D medical volume understanding. Photon directly operates on 3D scans and integrates Instruction-conditioned Token Scheduling (ITS) with Surrogate Gradient Propagation (SGP) to achieve adaptive and consistent token reduction during both training and inference. ITS discretely selects informative tokens in the forward path and predicts adaptive sample-specific thresholds from instruction-vision interactions. SGP ensures effective gradient back-propagation in a discrete token dropping scenario. Through this innovative design, Photon reduces computational and memory costs while preserving clinically critical information. Experiments on various medical visual question answering benchmarks demonstrate that Photon achieves state-of-the-art performance with substantially fewer tokens and less computational overhead.

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

# A APPENDIX

## A.1 METHODOLOGY

### A.1.1 TRAINING PROCESS

Unlike most previous 3D medical MLLM methods that use a three-stage pipeline (including vision pretraining, vision language alignment, and vision language instruction finetuning), we design a lighter and more efficient two-stage training strategy.

In Phase 1, we perform visual embedding alignment by training the 3D vision patch embedding layer of the visual encoder on volume-radiology report data with a learning rate of $1e^{-4}$ under annealing. The ViT backbone retains its pretrained weights from Qwen2.5-VL (Bai et al., 2025), and both the backbone and the MLP aligner are kept frozen to preserve the priors learned from large-scale natural images. The Qwen2.5-VL decoder is also frozen and serves only as the alignment target. We find that fully finetuning the vision encoder tends to cause overfitting and negatively affects the decoder's ability to follow instructions and interpret visual inputs. By updating only the 3D vision patch embedding layer, this stage improves efficiency, maintains generalization, and avoids the need to re-pretrain the visual encoder or re-align vision and language on large-scale medical data.

In Phase 2, based on the alignment obtained in Phase 1, we replace the Qwen2.5-VL language decoder with the Photon language decoder that integrates our ITS and SGP modules, while retaining the backbone parameters reused from Qwen2.5-VL. This design allows us to directly inherit the alignment results and the pretrained capabilities of Qwen2.5-VL without restarting pretraining from scratch. In this stage, we unfreeze all modules and finetune them on the task-specific training set for optimization with a learning rate of $1e^{-5}$ under a cosine annealing schedule.

### A.1.2 DETAILS OF THRESHOLD FEATURE CONSTRUCTION AND PRUNING

We detail the three feature blocks and the subsequent thresholding and selection.

**Normalization for ranking.** Given raw logits $d = \{d_j\}_{j \in \mathcal{V}^{(b)}}$, we define the min-max normalized saliency scores:

$$\rho_j = \frac{d_j - \min_{k \in \mathcal{V}^{(b)}} d_k}{\max_{k \in \mathcal{V}^{(b)}} d_k - \min_{k \in \mathcal{V}^{(b)}} d_k + \epsilon}, \qquad j \in \mathcal{V}^{(b)}.$$

This rescales logits to $[0, 1]$ and preserves within-sample ordering.

**Block 1: shape descriptor from normalized scores.** From $\rho = \{\rho_j\}$ we compute:

$$\Psi(\rho) = \big[\mu(\rho), \ \mathrm{Std}(\rho), \ \min(\rho), \ \max(\rho), \ Q_{1/4}(\rho), \ \mathrm{med}(\rho)\big] \in \mathbb{R}^6,$$

capturing central tendency, spread, range, lower quartile, and median in a scale-free manner.

**Block 2: scale descriptor from raw logits.** From $d$ we extract absolute-scale statistics:

$$\Phi(d) = \big[\mu(d), \ \mathrm{Std}(d), \ \min(d), \ \max(d)\big] \in \mathbb{R}^4,$$

which retain level and variability on the original linear scale.

**Block 3: compressed descriptor from signed log view.** To summarize large magnitudes while preserving sign, we use:

$$\mathrm{slog}(x) = \mathrm{sign}(x) \, \log\big(1 + |x|\big),$$

and define:

$$\Upsilon(\mathrm{slog}(d)) = \big[\mu(\mathrm{slog}(d)), \ \max(\mathrm{slog}(d))\big] \in \mathbb{R}^2.$$

This provides compressed indicators of overall level and dominant peaks.

### A.1.3 GRADIENT DYNAMICS OF THE SURROGATE

We provide a formal analysis of the surrogate gradient mechanism described in the main text. The token-level retention probability is defined as:

$$q = \sigma\left(\frac{\rho - \theta}{\tau_{\mathrm{ce}}}\right),$$

where $\rho$ denotes the normalized saliency score and $\theta$ is the instance-specific threshold predicted by ITP. The surrogate gradient with respect to $q$ takes the form:

$$\frac{\partial \mathcal{L}}{\partial q} = \beta \, d \, s \, \max\{q(1-q), \epsilon_{sat}\}, \qquad d = 0.5 - \mathrm{sg}(r), \quad r = \psi(z), \; \psi(0) = 0.5,$$

where $d$ is the directional term measuring deviation from the neutral point $0.5$, $s$ is the magnitude factor obtained from activation-gradient interactions, $\beta$ is a balancing coefficient, and the saturation term $\max\{q(1-q), \epsilon_{sat}\}$ prevents vanishing gradients near the extremes. As in the main text, all updates are restricted to visual positions, so that only active tokens contribute.

The partial derivatives of $q$ are:

$$\frac{\partial q}{\partial \theta} = -\frac{1}{\tau_{\mathrm{ce}}} \, \sigma'\left(\frac{\rho - \theta}{\tau_{\mathrm{ce}}}\right) < 0, \qquad \frac{\partial q}{\partial \rho} = \frac{1}{\tau_{\mathrm{ce}}} \, \sigma'\left(\frac{\rho - \theta}{\tau_{\mathrm{ce}}}\right) > 0,$$

which leads to:

$$\frac{\partial \mathcal{L}}{\partial \theta} = \frac{\partial \mathcal{L}}{\partial q} \cdot \frac{\partial q}{\partial \theta}, \qquad \frac{\partial \mathcal{L}}{\partial \rho} = \frac{\partial \mathcal{L}}{\partial q} \cdot \frac{\partial q}{\partial \rho}.$$

When the directional term satisfies $d < 0$, we obtain $\frac{\partial \mathcal{L}}{\partial q} < 0$. Consequently, $\frac{\partial \mathcal{L}}{\partial \theta} > 0$ and $\frac{\partial \mathcal{L}}{\partial \rho} < 0$, so gradient descent decreases $\theta$ and increases $\rho$. This enlarges the margin $\rho - \theta$, raises $q$, and thereby promotes token retention.

Conversely, when $d > 0$, we obtain $\frac{\partial \mathcal{L}}{\partial q} > 0$. In this case, $\frac{\partial \mathcal{L}}{\partial \theta} < 0$ and $\frac{\partial \mathcal{L}}{\partial \rho} > 0$, so gradient descent increases $\theta$ and decreases $\rho$. This reduces the margin $\rho - \theta$, lowers $q$, and thereby increases the tendency toward pruning.

Overall, the surrogate gradient dynamics remain consistent with the intended behavior: tokens assessed as informative are reinforced toward retention, while less informative tokens receive weaker gradients and are progressively pruned.

### A.1.4 Derivation of token importance via first-order approximation

Let $T_{\mathrm{vis}}^{\ell} \in \mathbb{R}^{L \times d}$ denote the visual hidden states at decoder layer $\ell$. Removing token $j$ is modeled as setting its hidden state $(T_{\mathrm{vis}}^{\ell})_j$ to zero while keeping all other tokens unchanged. Applying a first-order Taylor expansion of the training loss $\mathcal{L}$ around $T_{\mathrm{vis}}^{\ell}$ gives

$$\mathcal{L}(T_{\mathrm{vis}}^{\ell} - (T_{\mathrm{vis}}^{\ell})_j) \approx \mathcal{L}(T_{\mathrm{vis}}^{\ell}) - \left\langle (T_{\mathrm{vis}}^{\ell})_j, \left(\frac{\partial \mathcal{L}}{\partial T_{\mathrm{vis}}^{\ell}}\right)_j \right\rangle,$$

so that the predicted loss variation caused by discarding token $j$ is

$$\Delta \mathcal{L}_j \approx - \left\langle (T_{\mathrm{vis}}^{\ell})_j, \left(\frac{\partial \mathcal{L}}{\partial T_{\mathrm{vis}}^{\ell}}\right)_j \right\rangle.$$

This matches the Taylor-based importance criterion proposed for filter and neuron pruning (Molchanov et al., 2016). We therefore adopt the activation-gradient product as a token-level importance signal and define

$$\eta_j = \left\langle (T_{\mathrm{vis}}^{\ell})_j, \left(\frac{\partial \mathcal{L}}{\partial T_{\mathrm{vis}}^{\ell}}\right)_j \right\rangle,$$

This extends the classical Taylor criterion to multimodal token reduction with differentiable masking, and the resulting scores are standardized and clipped across active tokens for stable optimization.

## A.2 Experiments

### A.2.1 Experimental Setup

In our experiments, training was performed on 16 NVIDIA H20 GPUs with BF16 precision with Deepspeed ZeRO2 (Rasley et al., 2020), and inference was conducted on 8 NVIDIA H20 GPUs. We set $\tau_{\mathrm{ce}} = 0.2$ and configured SGP with $\epsilon_{\mathrm{std}} = 3 \times 10^{-3}$, $(s_{\min}, s_{\max}) = (0.5, 2.0)$, $\beta = 0.8$, $\epsilon_{sat} = 5 \times 10^{-3}$, $r_{min}$ and $r_{max}$ are set to 0.3 and 0.8 respectively. Unless otherwise noted, ITS and SGP are applied at $\ell = \ell_{n/4}$. Optimization uses AdamW, and FlashAttention2 (Dao, 2024) is enabled in all attention layers.

### A.2.2 TASKS, DATASETS AND METRICS

In Phase 1 of training, the visual embedding alignment process was trained on the volume-caption pairs of CT-RATE (we use 47,141 out of 50,188 pairs, reconstructed from 25,692 volumes belonging to 21,304 patients for training) (Hamamci et al., 2024b), AbdomenAtlas 3.0 (we use 8,334 out of 9,262 pairs for training) (Li et al., 2024a), and our internal dataset (we use 27,577 pairs for training). To address the potential overlap of identical volumes between Phase 1 and Phase 2, we adopt the Phase 2 data partitioning scheme in Phase 1, thereby preventing data leakage.

In Phase 2 of training, we finetune all model parameters using task-specific data. We employ two large-scale 3D CT VQA benchmarks: 3D-RAD (Gai et al., 2025) and DeepTumorVQA (Chen et al., 2025c), which together span descriptive, recognition, and reasoning-oriented diagnostic scenarios under clinically relevant protocols. We elaborate on these two datasets as follows.

3D-RAD (Gai et al., 2025) is a large-scale 3D medical VQA benchmark built upon CT-RATE (Hamamci et al., 2024b), comprising 16,188 CT scans from 11,255 patients with strict train-test separation. It covers six tasks (anomaly detection, image observation, medical measurement, existence detection, static temporal diagnosis, and longitudinal temporal diagnosis), supporting both open- and closed-ended questions. QA pairs are generated from radiology reports and multi-label annotations using task-specific templates, and refined through an LLM with human verification pipeline with over 91% factual alignment. The final dataset provides 33,910 curated benchmark QA pairs and 136,195 training pairs, enabling comprehensive evaluation of multi-task and multi-temporal reasoning in clinically realistic settings.

DeepTumorVQA (Chen et al., 2025c) is built from 9,262 abdominal CT volumes. The benchmark provides 355,962 training and 39,650 testing QA pairs across measurement, recognition, visual reasoning, and medical reasoning categories, covering both multiple-choice and free-text formats. Evaluation follows task-specific protocols, including accuracy for multiple-choice, exact match for free-text answers, mean relative accuracy (MRA) (Yang et al., 2025c) for numerical prediction, and CE for region-level clinical correctness. This design ensures reliable assessment of both reasoning capability and clinical validity.

SLAKE (Liu et al., 2021) is a large-scale bilingual medical visual question-answering benchmark that features richly annotated radiology images and knowledge graph support. The dataset covers multiple imaging modalities and several body regions. QA pairs span question types such as imaging plane, modality, organ detection, abnormalities, and knowledge-graph reasoning. The benchmark supports both closed-ended and open-ended question formats and is designed for evaluating visual reasoning plus medical knowledge reasoning in a multilingual setting.

NExT-QA (Xiao et al., 2021) is a video question answering benchmark designed to move beyond scene description toward explaining causal and temporal action relations. It contains 5,440 real-world videos and 52,044 human-annotated QA pairs, organized into three categories: causal questions that require identifying visible cause–effect relations, temporal questions that probe ordering of actions, and descriptive questions covering objects, attributes and events.

MedFrameQA (Yu et al., 2025) is a benchmark designed for multi-image medical visual question answering that reflects more realistic clinical workflows. MedFrameQA contains 2,851 clinically grounded VQA items constructed from 9,237 high-quality frames extracted from 3,420 instructional medical videos. Each question is paired with two to five temporally coherent frames that share a continuous diagnostic focus. The dataset spans nine human body systems and 43 organs across diverse imaging modalities, forming a challenging setting that requires models to integrate complementary evidence across multiple images rather than relying on single-frame cues.

We evaluate model performance using a combination of text generation and classification metrics. For free-text tasks, we adopt BLEU, ROUGE, and BERT Score to assess fluency, lexical overlap, and semantic similarity between generated and reference answers. BLEU captures $n$-gram precision, ROUGE emphasizes recall-oriented overlap, and BERT Score leverages contextual embeddings for semantic alignment. For categorical predictions such as existence detection and temporal diagnosis, Accuracy is reported. For numerical free-text answers, we further apply the Mean Relative Accuracy (MRA) (Yang et al., 2025c) metric, which measures correctness under a range of relative error thresholds, reflecting clinical tolerance in quantitative assessment. Together, these metrics provide a comprehensive evaluation across descriptive, reasoning, and measurement-oriented tasks.

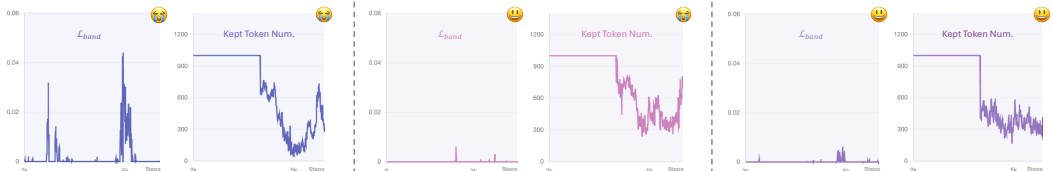

Figure 5: Visualization of retention band triggers (Eq. (18)) and the number of kept tokens. Left: training without robust regularization, middle: training without flip regularization, right: training with both regularizations. A lower activation frequency and smaller magnitude of the retention band indicate more stable pruning during training.

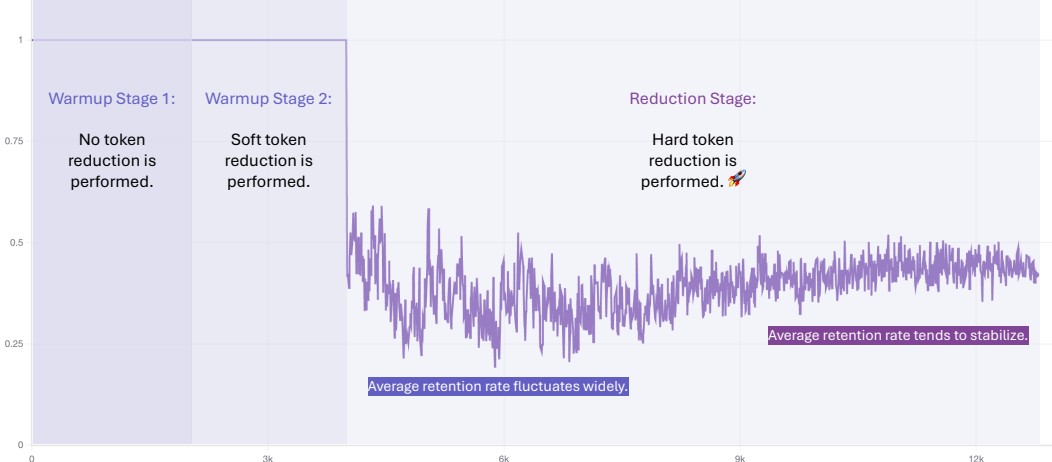

Figure 6: Visualization of retention rate of training time. Our token reduction method exhibits an average retention rate that fluctuates within a reasonable range before steadily stabilizing, indicating convergence toward an appropriate per-sample threshold estimation.

### A.2.3 ANALYSIS OF SOFT RETENTION BAND AND REGULARIZATION

The soft retention band is designed as a safeguard mechanism rather than a regular penalty. Under normal conditions, it has little impact on performance because most retention probabilities lie within the expected range and are not penalized. Its main role emerges in extreme situations, where the predictor deviates sharply from a reasonable retention ratio. In such cases, the band introduces corrective pressure that pulls the retention ratio back into a stable range, preventing collapse.

Fig. 5 compares the trigger visualization of the soft retention band and the number of kept tokens under different regularization settings. The instability mainly arises from the limited visual priors of the pretrained backbone in 3D medical domains. Without sufficient prior knowledge, the predictor struggles to identify clinically relevant regions and systematically underestimates the importance of visual tokens. As shown in the left plots, in the absence of the robust regularizer this bias is amplified: the predictor tends to discard visual tokens, leading to unstable and sometimes extreme fluctuations in the retention ratio that reflect insufficient grounding in image evidence. The middle plots show that when only the robust term is applied but the flip regularizer is removed, pruning becomes smoother but the model starts to overfit spurious retention patterns, which undermines stability. By contrast, the right plots demonstrate that using both regularizers produces stable retention dynamics: the robust term suppresses neglect of visual content and constrains excessive pruning, while the flip term prevents shortcut reliance on fixed masks. Together, they enforce a balanced reliance on both visual and textual signals, resulting in smoother training and stronger visual grounding.

### A.2.4 TOKEN REDUCTION ANALYSIS

Fig. 6 illustrates the progression of the average token retention rate during training Phase 2. In Warmup Stage 1, no reduction is applied and the rate remains fixed at 1.0. In Warmup Stage 2, the retention rate also stays at 1.0, but soft masking is applied to tokens so that the predictor begins learning threshold estimation, providing a gradual transition between full retention and hard pruning.

Table 5: Clinical Metrics of Kidney tumor recognition (sensitivity, specificity, and accuracy (%)).

| Metric | RadFM | M3D-P3 | M3D-L2 | Merlin | CT-CHAT | Photon | Photon Bootstrap |
|---|---|---|---|---|---|---|---|
| Sensitivity | 75.2 | 0.0 | 0.0 | 50.2 | 0.0 | 81.2± 1.7 | 81.7 [78.6, 84.7] |
| Specificity | 98.6 | 100.0 | 100.0 | 51.2 | 100.0 | 95.2± 0.1 | 95.3 [93.8, 96.6] |
| Accuracy | 88.6 | 57.4 | 57.4 | 50.7 | 57.4 | 89.3± 0.7 | 89.5 [87.9, 91.1] |

Table 6: Photon performance on the AbdomenAtlas3.0-style OOD split of DeepTumorVQA. Results are reported for multi-choice and free-text variants (accuracy (%)).

| Method | Avg. keep tok. | Meas. | Recog. | Vis. Rsn. | Med. Rsn. | Overall |
|---|---|---|---|---|---|---|
| Photon OOD Multi-Choice | 0.34K | 0.745 | 0.599 | 0.636 | 0.677 | 0.648 |
| Photon OOD Free-Text | 0.40K | 0.459 | 0.643 | 0.520 | 0.599 | 0.570 |

Once entering the Reduction Stage, hard pruning is applied: the retention rate first oscillates as the predictor adapts and then gradually stabilizes. This staged schedule prevents premature pruning and enables a smooth shift from probability learning to discrete token removal.

### A.2.5 ANALYSIS IN CLINICAL METRICS

As noted in (Chen et al., 2025c), lesion recognition poses greater challenges for MLLMs than general VQA tasks, since reliable clinical application requires balanced sensitivity, specificity, and accuracy. Table 5 highlights the diversity of model behaviors: some approaches emphasize specificity at the expense of detection, while others show higher sensitivity but limited overall stability. These differences reflect the difficulty of consistently capturing subtle lesion signals in raw 3D volumes without explicit spatial supervision.

To account for randomness, we repeat the experiment with ten independent seeds on under the same protocol. We report mean ± standard deviation for all medical metrics. Photon exceeds the strongest baseline under every seed, which indicates that the improvements are stable with respect to initialization. For the main Photon results, we also estimate test-set uncertainty with 1000 bootstrap resamples of evaluation questions and report 95 percent confidence intervals. The intervals remain narrow, showing that the observed gains are not due to sampling noise.

In addition, Photon achieves a more stable balance across the three metrics. It maintains competitive accuracy while improving lesion detection rates, thereby reducing the likelihood of missed cases without compromising overall reliability. This balance across sensitivity, specificity, and accuracy suggests that Photon better aligns with clinical requirements compared with prior MLLM baselines.

### A.2.6 ZERO-SHOT AND OUT-OF-DOMAIN EVALUATION

In this section we study whether Photon maintains robustness and efficiency when evaluated outside its training distribution. Here we focus on two complementary scenarios: (1) institution-level out-of-domain (OOD) evaluation on DeepTumorVQA (Chen et al., 2025c) using the AbdomenAtlas3.0-based (Li et al., 2024a) split, and (2) a fully training-free zero-shot setting on MedFrameQA (Yu et al., 2025).

**Institution-level OOD split on DeepTumorVQA.** To evaluate robustness under distribution shift, we follow the AbdomenAtlas3.0 OOD split and redivide DeepTumorVQA into training and test sets based on medical institutions, which also induces changes in population, devices, and imaging protocols. We report results on free-text and multi-choice variants. Table 6 summarizes task-wise performance and the average number of retained vision tokens per sample.

Compared with the in-domain results reported in Table 2, the OOD setting leads to a moderate drop of approximately 0.038 and 0.049, respectively. This indicates that Photon preserves most of its accuracy under institution-level distribution shift while still retaining only $0.34K \sim 0.40K$ visual tokens on average, which confirms that the pruning mechanism does not destabilize cross-domain generalization.

**Training-free zero-shot setting on MedFrameQA.** To further isolate the effect of pruning and threshold estimation, we evaluate a training-free variant of Photon where the token scheduling module

Table 7: Zero-shot results on full MedFrameQA dataset. No additional medical fine-tuning is performed (accuracy (%)).

| Method | Avg. keep tok. | CT | MRI | X-ray | Other | Ultrasound | Overall |
|---|---|---|---|---|---|---|---|
| Qwen2.5-VL 3B Zero-Shot | 7.00K | 0.461 | 0.460 | 0.485 | 0.477 | 0.432 | 0.460 |
| Qwen2.5-VL 3B + ITS Zero-Shot | 3.28K | 0.459 | 0.456 | 0.488 | 0.470 | 0.426 | 0.457 |

is attached to Qwen2.5-VL 3B without any additional medical fine-tuning. We test this configuration in a strict zero-shot setting on MedFrameQA and compare it with the plain Qwen2.5-VL 3B backbone. Results across modalities and the average token keep rate are shown in Table 7.

The training-free Photon variant reduces the average token keep rate from $7.00K$ to $3.28K$ while maintaining almost identical zero-shot accuracy. Since the token scheduling module is attached as a plug-in and operates independently from the MLLM backbone parameters, these results indicate that the pruning and threshold estimation strategy preserves the backbone's zero-shot and generalization ability, even when no task-specific fine-tuning is applied.

### A.2.7 MORE ABLATION STUDY

**Hyperparameters affecting pruning.** We test the three hyperparameters that most directly control pruning behavior: the temperature $\tau_{\text{ce}}$, the surrogate gradient weight $\beta$, and the saturation threshold $\epsilon_{\text{sat}}$. As summarized in Table 8, changing these values mainly adjusts the average number of retained tokens, while task accuracy on all three 3D-RAD subtasks and the generative metrics remains stable. This supports the claim that our default configuration is already in a robust regime and that aggressive hyperparameter tuning is unnecessary in practice.

**Robustness to noisy thresholds.** To probe the robustness of the threshold predictor, we perturb the predicted thresholds at inference time with random multiplicative noise in $[-10\%, +10\%]$. The noisy variant in Table 8 shows only mild performance degradation, indicating that Photon does not rely on finely tuned thresholds and is tolerant to moderate estimation errors.

**Scaling across backbones.** Finally, we examine cross-model scaling. On Qwen2.5-VL, extending Photon from a 3B to a 7B backbone improves performance across tasks with similar retention ratios, and attaching Photon to Qwen3-VL (Yang et al., 2025a) 2B yields consistent pruning behavior without training collapse. Together, these results show that Photon is stable under hyperparameter changes, robust to noisy thresholds, and transferable across different backbone sizes and architectures.

### A.3 EXTEND PHOTON TO MORE MODALITIES

**Instruction-conditioned scheduling across modalities.** Instruction-conditioned Token Scheduling (ITS) does not operate on raw intensities or fixed spatial coordinates. Instead, it predicts a retention ratio from a normalized alignment distribution produced by the instruction-visual cross-attention. This distribution is normalized within each sample and depends on the relative ordering of token importance, rather than absolute intensities or fixed noise patterns. Therefore, ITS can adapt to changes in imaging centers, scanning protocols, slice numbers, orientations, spacing, and noise levels.

**Results on MedFrameQA (Yu et al., 2025) and SLAKE (Liu et al., 2021).** To test this property, we apply Photon to MedFrameQA, which contains diverse modalities and variable frame counts, slice numbers, and resolutions from multiple sources. We randomly split MedFrameQA (Yu et al., 2025) into training and testing sets with a $9:1$ ratio, and compare the Qwen2.5-VL 3B backbone with and without Photon. As shown in Table 9, Photon reduces the average token keep rate from $1.0$ to about $0.43$ while slightly improving the overall accuracy, with consistent gains on CT, MRI, and X-ray and no degradation on ultrasound. This suggests that the same ITS configuration that works for 3D-RAD (Gai et al., 2025) and DeepTumorVQA (Chen et al., 2025c) also remains stable on a heterogeneous multimodal benchmark, supporting the view that ITS learns a task-guided token importance structure instead of a dataset-specific mask pattern.

We also study transfer to MRI data using the SLAKE (Liu et al., 2021) dataset. Here we focus on MRI-based VQA in SLAKE (Liu et al., 2021) and again compare the non-pruned Qwen2.5-VL 3B baseline with our pruned variant. Photon cuts the average keep rate to about $0.38$ while keeping

Table 8: Ablation on key hyperparameters and robustness of Photon. $\tau_{ce}$ controls the temperature for contrastive estimation; $\beta$ balances surrogate gradient strength; $\epsilon_{sat}$ defines the saturation suppression threshold. The last block tests robustness under noisy threshold perturbation and cross-model scaling.

| Setting | E.D. Acc. | S.T.D. Acc. | L.T.D. Acc. | Medical Measurement BLEU | ROUGE | BERT | Image Observation BLEU | ROUGE | BERT | Anomaly Detection BLEU | ROUGE | BERT | Avg. Train Tok. |
|---|---|---|---|---|---|---|---|---|---|---|---|---|---|
| *Effect of $\tau_{ce}$ (temperature for contrastive estimation).* | | | | | | | | | | | | | |
| $\tau_{ce}$=0.10 | 82.44 | 51.66 | 77.66 | 38.65 | 39.87 | 96.20 | 51.11 | 56.77 | 93.61 | 41.95 | 46.91 | 91.86 | 0.42K |
| $\tau_{ce}$=0.20 | 83.07 | 52.86 | 77.01 | 37.74 | 39.36 | 96.14 | 51.59 | 56.66 | 93.62 | 42.33 | 47.50 | 91.96 | 0.39K |
| $\tau_{ce}$=0.50 | 82.88 | 52.13 | 77.71 | 38.59 | 39.77 | 96.25 | 51.61 | 56.16 | 93.53 | 41.73 | 46.81 | 91.87 | 0.32K |
| *Effect of $\beta$ (surrogate gradient weight).* | | | | | | | | | | | | | |
| $\beta$=0.5 | 82.99 | 52.25 | 75.57 | 37.05 | 38.45 | 96.04 | 51.60 | 56.12 | 93.64 | 41.76 | 47.18 | 91.90 | 0.46K |
| $\beta$=0.8 | 83.07 | 52.86 | 77.01 | 37.74 | 39.36 | 96.14 | 51.59 | 56.66 | 93.62 | 42.33 | 47.50 | 91.96 | 0.39K |
| $\beta$=1.0 | 82.70 | 52.86 | 76.63 | 38.51 | 39.70 | 96.11 | 52.08 | 56.79 | 93.72 | 41.99 | 47.52 | 91.93 | 0.40K |
| *Effect of $\epsilon_{sat}$ (saturation suppression threshold).* | | | | | | | | | | | | | |
| $\epsilon_{sat}$=1e$^{-3}$ | 82.87 | 51.53 | 76.78 | 37.09 | 38.29 | 96.00 | 52.14 | 56.72 | 93.66 | 42.20 | 47.22 | 91.97 | 0.45K |
| $\epsilon_{sat}$=5e$^{-3}$ | 83.07 | 52.86 | 77.01 | 37.74 | 39.36 | 96.14 | 51.59 | 56.66 | 93.62 | 42.33 | 47.50 | 91.96 | 0.39K |
| $\epsilon_{sat}$=1e$^{-2}$ | 82.83 | 53.27 | 76.40 | 37.67 | 39.10 | 96.20 | 51.27 | 56.11 | 93.65 | 41.78 | 47.08 | 91.91 | 0.61K |
| *Robustness under noisy threshold perturbation.* | | | | | | | | | | | | | |
| W/o Noise | 83.07 | 52.86 | 77.01 | 37.74 | 39.36 | 96.14 | 51.59 | 56.66 | 93.62 | 42.33 | 47.50 | 91.96 | — |
| [−10%, +10%] | 82.87 | 52.33 | 76.59 | 37.63 | 39.05 | 96.13 | 51.43 | 56.72 | 93.61 | 42.15 | 47.38 | 91.93 | — |
| *Cross-model scaling: Photon with different backbone and sizes.* | | | | | | | | | | | | | |
| Photon (Q2.5VL 3B) | 83.07 | 52.86 | 77.01 | 37.74 | 39.36 | 96.14 | 51.59 | 56.66 | 93.62 | 42.33 | 47.50 | 91.96 | 0.39K |
| Photon (Q2.5VL 7B) | 83.50 | 53.90 | 78.03 | 40.70 | 41.90 | 96.20 | 53.34 | 57.65 | 93.77 | 42.66 | 48.14 | 91.99 | 0.41K |
| Photon (Q3VL 2B) | 82.28 | 52.21 | 75.03 | 37.65 | 39.22 | 95.84 | 50.59 | 55.30 | 93.40 | 41.30 | 46.98 | 91.84 | 0.37K |

Table 9: Token retention and performance on MedFrameQA and SLAKE (accuracy).

| Method | MedFrameQA Avg. keep rate | CT | MRI | X-ray | Other | Ultrasound | Avg. Acc. | SLAKE Avg. keep | Acc. | Match / Total |
|---|---|---|---|---|---|---|---|---|---|---|
| Qwen2.5VL 3B | 1.000 | 0.744 | 0.745 | 0.893 | 0.667 | 0.486 | 0.727 | 1.000 | 0.8446 | 413/489 |
| Ours 3B | 0.432 | 0.761 | 0.787 | 0.857 | 0.750 | 0.486 | 0.745 | 0.375 | 0.8425 | 412/489 |

accuracy almost unchanged, with only one fewer correct case out of 489. This indicates that the pruning strategy is not tied to CT-specific characteristics and can be applied to other medical imaging modalities.

**Extension to video reasoning.** Photon can also be applied to video reasoning tasks. In this setting, the base Qwen2.5-VL model is already pre-trained on large-scale visual data, so we simplify the training schedule: the two warmup stages and robustness regularization used for 3D medical training are removed, and the pruning module is trained directly with the main task loss.

To demonstrate this, we apply Photon to the NExT-QA (Xiao et al., 2021) dataset and compare it with the non-pruned Qwen2.5-VL 3B baseline. Table 10 shows that Photon reduces the average token keep rate by more than half while keeping the overall accuracy essentially unchanged. Across causal, descriptive, and temporal question types, the differences remain small, and several categories show slight improvements. This confirms that ITS and SGP can be used in video understanding settings with minimal modifications, providing substantial token reduction without sacrificing performance.

Across CT, MRI, multimodal medical frames, and videos, Photon consistently cuts visual tokens by more than half while keeping accuracy comparable to non-pruned baselines. This shows that the instruction-conditioned pruning design is not restricted to a single imaging protocol or task format and can generalize across diverse modalities and benchmarks.

### A.3.1 More visualization results and failure cases.

**Qualitative behavior and typical failures.** Fig. 7 presents additional visualization results, including both correct predictions and failure cases. Photon successfully identifies major findings such as air bronchogram signs and kidney tumors, but limitations appear in more subtle scenarios. For example, when estimating the size of a nodule on the minor fissure, the model underestimates the measurement (4×3 mm vs. ground-truth 5×3 mm), which reflects the difficulty of precise quantitative reasoning

Table 10: Token retention and performance on NExT-QA (accuracy).

| Method | Avg. keep rate | Question type accuracy | | | | | | | | Avg. Acc |
| | | Causal-How | Causal-Why | Desc.-Count | Desc.-Loc | Desc.-Other | Temp.-Now | Temp.-Next | Temp.-Present | |
|---|---|---|---|---|---|---|---|---|---|---|
| Qwen2.5VL 3B | 1.000 | 0.7545 | 0.7738 | 0.6429 | 0.9358 | 0.8533 | 0.7760 | 0.7269 | 0.7097 | 0.7729 |
| Ours 3B | 0.473 | 0.7502 | 0.7693 | 0.6522 | 0.9358 | 0.8483 | 0.7631 | 0.7455 | 0.7527 | 0.7723 |

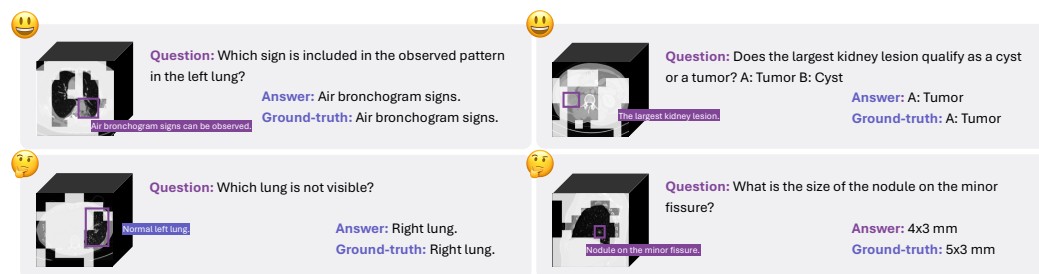

Figure 7: More visualization results and failure cases by Photon.

on small targets. In the lung visibility case, although the prediction is correct, the pruning behavior removes the invisible lung and retains the normal one. This likely arises because cases with invisible lung fields are rare in the training data, so the model tends to assign higher importance to common anatomical regions. These examples indicate that errors mostly occur in fine-grained quantitative estimation and rare structural patterns.

**Type I: small lesion measurements.** The first type of failure concerns small lesion measurements, which are inherently more challenging than tasks such as existence detection. The difficulty comes from several factors: the number of annotated cases for tiny lesions is limited, annotations are not fully standardized and show inter-rater variability, and continuous quantities must be expressed using discrete words. As a result, all methods perform worse on measurement than on existence detection. However, Photon still outperforms or is at least comparable to non-pruned baselines on measurement, which suggests that these errors are mainly due to task and supervision limitations rather than instability introduced by token pruning.

**Type II: rare or anatomically unusual cases.** The second type of failure appears in rare or anatomically unusual cases, where long-tail samples in training make the model rely more strongly on common structures and become more uncertain in rare regions. For example, in the 3D-RAD (Gai et al., 2025) existence detection task, compared with more than one hundred thousand cases in total, some findings only have a few hundred positive case. Such imbalance makes it harder for the model to assign high saliency to uncommon structures, so both attention and pruning can be less reliable in these regions, even when the backbone itself remains stable.

### A.3.2 STATISTICAL ANALYSIS ON TOKEN PRUNING

**Task-conditioned token retention.** To address whether token dropping depends on instructions, we analyze how task types change retention for the same organ. For lung-related instructions in 3D-RAD, Table 11 reports the average number of retained lung tokens for Image Observation, Anomaly Detection, Medical Measurement, and Existence Detection. Medical Measurement keeps the most tokens, while Existence Detection keeps the fewest. This pattern shows that the retention ratio is not a fixed global rate, but is systematically adjusted according to task type, which supports that pruning is conditioned on the instruction and task semantics.

**Spatial dissimilarity across instructions.** We further study how the spatial distribution of retained regions changes with different organ-related instructions. We measure Jaccard dissimilarity (defined as 1 minus the overlap) between retained-region masks obtained under different prompts. The right part of Table 11 gives dissimilarity values between organ pairs, which fall in the range 0.52–0.69. These values indicate noticeable but structured changes in the retained regions when the organ in the instruction changes, again suggesting that Photon's pruning responds to semantic differences instead of acting as a fixed spatial mask.

Table 11: Task-conditioned lung token retention (left) and organ-wise Jaccard dissimilarity (right).

| Task | Image observation | Anomaly detection | Pair | kidney vs liver | liver vs spleen | kidney vs lung |
|---|---|---|---|---|---|---|
| Avg. retained tokens | 485.3 | 479.6 | Dissimilarity | 0.6810 | 0.6646 | 0.5530 |
| Task | Medical measurement | Existence detection | Pair | liver vs lung | lung vs spleen | kidney vs spleen |
| Avg. retained tokens | 499.5 | 455.9 | Dissimilarity | 0.6852 | 0.5237 | 0.5247 |

Table 12: Block-level lung coverage for Photon (3B) on lung-related instructions in 3D-RAD.

| Method | Global coverage | Mean per volume coverage |
|---|---|---|
| Ours 3B | 0.831 | 0.818 |

**Overlap with lung segmentations.** To check whether instruction-conditioned pruning still preserves key anatomical regions, we extract all lung-related instructions from 3D-RAD and perform a block-level overlap analysis between retained tokens and reference lung segmentations on 3D volumes. Pre-trained segmentation models provide the reference lung masks, which are resampled to the token grid used by the pruning module. For each volume, we compute the fraction of lung-positive blocks that are retained and then aggregate these statistics across the lung subset. As shown in Table 12, Photon retains a large fraction of lung parenchyma blocks with consistent coverage across volumes. Given that some lesion-specific queries do not require observing the entire lung parenchyma, this degree of coverage is reasonable. Overall, these visualization and quantitative analyses indicate that Photon's token scheduling is instruction-conditioned: both the number and spatial distribution of retained tokens adapt to the task and target organ.

## A.4 LIMITATIONS AND FUTURE WORKS

Although our method demonstrates robust empirical performance, prospective clinical validation and human-centric user studies with medical practitioners remain vital for future work. A promising avenue for research involves integrating temporal and longitudinal imaging to facilitate the coherent modeling of disease progression (Wang et al., 2025b;a). Future studies could also investigate the incorporation of reinforcement learning (Jiang et al., 2026; Chen et al., 2025a; Lin et al., 2025c;b) or self-supervised objectives to enable more adaptive token utilization. Furthermore, enhancing computational efficiency through refined KV-Cache management (Wang et al., 2026b) and more inference techniques (Ma et al., 2025) represents a critical direction for further exploration.

Beyond VQA tasks, Photon's utility could be extended to a broader spectrum of clinical applications, including cross-institutional generalization, interactive diagnostic support (Zhang et al., 2025b), and integration into large-scale multimodal pretraining pipelines (Liu et al., 2025a) to accommodate diverse modalities (Fang et al., 2025; Li et al., 2023c; 2025b; Fang et al., 2024). Integrating auxiliary diffusion modules Wang et al. (2026a); Chen et al. (2025b) presents a compelling opportunity to generate more intuitive and interpretable visual results. Combining more tasks like segmentation He et al. (2025); Liu et al. (2025c;d); Wang et al. (2025c; 2024a) is also worth exploring.

## A.5 ETHICS STATEMENT

This study was conducted in accordance with ethical standards for medical data usage. We obtained authorization to use the DeepTumorVQA (Chen et al., 2025c) and AbdomenAtlas 3.0 datasets (Li et al., 2024a). For the 3D-RAD (Gai et al., 2025) dataset, which is derived from CT-RATE (Hamamci et al., 2024b), we followed the CC BY-NC-SA license under which it is released. All datasets used in this work, whether publicly available or private, were fully de-identified to protect patient privacy. For data requiring explicit authorization, formal permission was obtained for use.

## A.6 REPRODUCIBILITY STATEMENT

To support reproducibility, we provide detailed descriptions of the model architecture, training objectives, training procedures, and optimization settings in the main text and appendix. Experimental configurations, including hyperparameters, evaluation metrics, and implementation details, are

documented to enable replication. The relevant source code and evaluation agreements will be released to the community after undergoing institutional review and authorization.

### A.7 THE STATEMENTS OF USING LARGE LANGUAGE MODELS

In the preparation of this manuscript, large language models were employed for language polishing and grammar correction. All scientific content, data analyses, results, and conclusions were conceived, conducted, and verified by the authors. The use of LLMs was limited to improving readability and ensuring grammatical accuracy, without generating or altering any substantive scientific material.

## B ACKNOWLEDGEMENT

This work was supported by the STI 2030-Major Projects under Grant 2021ZD0201404 and Shenzhen Key Laboratory of New Generation Interactive Media Technology Innovation (ZDSYS20210623092001004). This work was also supported by Alibaba Group through Alibaba Research Intern Program.

