# OpenReview forum: "Photon: Speedup Volume Understanding with Efficient Multimodal Large Language Models"
_ICLR.cc/2026/Conference — ICLR 2026 Poster_

### Official Review · Reviewer_TVyY · 2025-10-15

**Soundness:** 3
**Presentation:** 3
**Contribution:** 2
**Rating:** 4
**Confidence:** 4

**Summary:**

This paper introduces Photon, an efficient multimodal large language model (MLLM) framework designed for 3D medical volume understanding—particularly in clinical visual question answering (VQA) tasks. Existing MLLMs struggle with 3D data because of high computational costs and loss of spatial continuity from slice-based or fixed-length token compression methods. Photon addresses these limitations by introducing a variable-length token representation and adaptive token reduction strategies that preserve volumetric fidelity while reducing computational load.

**Strengths:**

The proposed Instruction-conditioned Token Scheduling (ITS) and Surrogate Gradient Propagation (SGP) mechanisms are well presented.
The model achieves strong empirical performance on challenging benchmarks 3D-RAD and DeepTumorVQA, including tasks that require fine-grained spatial reasoning and longitudinal understanding—domains where many prior vision-language models underperform. Photon achieves significant reductions in training and inference memory (up to 5× speedup, 2–3× memory reduction) while preserving or improving task performance. The ablation study does clearly show that each component contributes to overall performance and training efficiency.

**Weaknesses:**

1. The technical novelty remains questionable. Visual token reduction (adaptive or non-adaptive via instruction tokens) has already been established in many previous papers. For instance, ATP-LLaVA [1] learns adaptive thresholds and drops tokens per instance/layer. IVTP also perform instruction-conditioned token selection with adaptive thresholds [2]. TransPrune [3] likewise uses instruction-guided attention to prune, so the core idea of instruction-aware and adaptive token pruning is not new. What Photon adds is mainly the 3D medical-volume setting.

2. Limited theoretical guarantees on convergence. While the surrogate gradient propagation in Section 3.3 and Appendix A.1.3 shows local consistency, this only guarantees alignment of update direction, not convergence or optimality. There is no formal guarantee that the retained set of tokens optimizes any global objective or converges to an optimal pruning schedule. A proof that shows the surrogate gradient descent converges to a fixed point would be interesting.

3. The experiments are limited to CT data. While the paper suggests extensibility to other modalities (MRI), no results or even small-scale demonstrations support these claims.

4. While the paper shows failure analysis, no standard deviation or confidence interval is shown. Can the authors quantify using statistical tests to show whether gains are significant?

References:

1. https://openaccess.thecvf.com/content/CVPR2025/papers/Ye_ATP-LLaVA_Adaptive_Token_Pruning_for_Large_Vision_Language_Models_CVPR_2025_paper.pdf

2. https://www.ecva.net/papers/eccv_2024/papers_ECCV/papers/02577.pdf

3. https://arxiv.org/abs/2507.20630

4. https://arxiv.org/abs/2403.17834

**Questions:**

Although the paper promises code release, there is no current code repository or full details sufficient for reproduction (e.g., dataset loading, data processing, hyperparameter details). Will the authors release the code?

How robust is the model to misestimated token importance or noisy retention thresholds? The current surrogate gradient strategy assumes reasonably accurate saliency estimates and threshold predictions. In practice, misestimation may lead to dropping critical visual tokens, especially early in training. Did the authors evaluate robustness under saliency noise or threshold perturbations (e.g., through ablation or noisy injection)? What prevents collapse into degenerate token usage?

How do retained tokens spatially distribute in 3D volumes, and are critical structures preserved? The paper provides visualization of retained regions, but does not evaluate whether anatomically or clinically critical regions (e.g., organs, tumors) are consistently retained across examples. Have the authors considered overlap analysis with reference segmentations or bounding boxes to quantify spatial alignment between retained tokens and clinically important areas?

---

> ### Author Response · Authors · 2025-11-22
> **Response to Reviewer TVyY (1/4)**
>
> We sincerely thank the reviewer for taking the time to evaluate our work, for the valuable suggestions, and for recognizing the design of ITS and SGP, the strong performance on 3D-RAD and DeepTumorVQA, the efficiency gains, and the contributions demonstrated by our ablation study.
>
> **W1 Analysis of technical novelty.**
>
> **Photon is not a direct adaptation of existing pruning methods to 3D medical tasks.** Although it also uses vision-language attention scores, Photon introduces instruction-guided dynamic pruning and learnable threshold prediction, enabling instruction-dependent retention without manual thresholds. Unlike prior methods designed mainly for inference-time speedup, Photon applies a consistent pruning strategy in both training and inference, achieving real acceleration and stable optimization.
>
> **In comparison**, ATP-LLaVA [1] performs dynamic pruning but still constrains the final retention rate and uses soft masking during training, limiting training-time acceleration. IVTP [2] allows adaptive pruning ratios per sample but requires manual threshold selection and introduces additional CLIP Text Tokenizer, which reduces the efficiency. TransPrune [3] uses a fixed pruning-rate strategy and is not designed for the training phase. VisionZip [4] offers training-time pruning but relies on manually selecting a global retention rate.
>
> We have discussed **ATP-LLaVA [1]**, **IVTP [2]**, and **TransPrune [3]** in related works. However, as they are not open-sourced, we instead adapted and evaluated other highly-related pruning approaches **VisionZip [4]** and **HiPrune [5]** in 3D medical scenario.
>
> Since these methods do not provide case-specific adaptive retention guided by instruction-related visual content, we evaluate them at fixed 30%, 50%, and 70% retention ratios on 3D-RAD [6]. These methods were not designed for training-time acceleration or stability, and many lack training protocols. In 3D medical tasks, where the base model has limited prior knowledge, applying them during training can remove essential tokens too early and cause language-only degeneration. For fairness, we train the full Qwen2.5-VL model without pruning to ensure proper optimization, and evaluate their pruning strategies using the official code based solely on prediction performance.
>
> |Methods(Keep Vision Token Per Sample)|Existence Detection|Static Temporal Diagnosis|Longitudinal Temporal Diagnosis|Medical Measurement|Image Observation|Anomaly Detection|Infer Speed|
> |-|-|-|-|-|-|-|-|
> |Qwen2.5 VL (7.0K)|81.97|47.62|75.36|37.00/38.57/96.04|52.72/56.48/93.63|42.01/47.36/91.93|2.30 Tok/s|
> |VisionZip (2.1K)|82.00|47.19|75.42|37.04/38.69/96.06|52.51/56.36/93.63|41.96/47.39/91.93|2.32 Tok/s|
> |VisionZip (3.5K)|81.99|47.74|75.93|37.11/38.68/96.07|52.72/56.60/93.65|41.98/47.36/91.92|2.23 Tok/s|
> |VisionZip (4.9K)|82.01|47.40|75.88|37.07/38.82/96.04|52.57/56.43/93.64|42.01/47.39/91.93|2.18 Tok/s|
> |HiPrune (2.1K)|81.99|48.08|75.50|37.00/38.59/96.05|52.56/56.46/93.64|42.08/47.32/91.93|0.76 Tok/s|
> |HiPrune (3.5K)|81.97|47.84|75.58|37.14/38.75/96.07|52.65/56.60/93.63|41.92/47.31/91.91|0.75 Tok/s|
> |HiPrune (4.9K)|82.02|47.19|75.35|37.00/38.62/96.06|52.57/56.44/93.62|41.94/47.40/91.91|0.73 Tok/s|
> |Photon (Dynamic)|83.07|52.86|77.01|37.74/39.36/96.14|51.59/56.66/93.62|42.33/47.50/91.96|4.12 Tok/s|
> |Photon MAX (Dynamic)|85.50|57.79|79.70|39.44/40.96/96.17|52.83/58.72/93.70|42.85/48.89/92.00|2.60 Tok/s|
>
>
> Most previous pruning methods, including VisionZip [4] and HiPrune [5], are not compatible with Flash Attention in the visual encoder and require full attention computation across all encoder layers, which causes clear efficiency losses. In our setting with large visual token counts and short answers, they do not provide prediction-speed gains and reduce accuracy. Our method improves both accuracy and inference speed.

---

> ### Author Response · Authors · 2025-11-22
> **Response to Reviewer TVyY (2/4)**
>
> **W2 Theoretical guarantees on convergence.**
>
> We thank the reviewer for raising the concern on convergence. In Section 3.3 and Appendix A.1.3, the threshold estimator's training signal is solely derived from the primary task loss, so that any change in pruning is driven directly by the main objective under the soft retention band regularization. For clarity, we present the following analysis at the level of a single visual token and omit the token index. To be specific, we relax the discrete keep decision into probabilities:
>
> $q_j = \sigma\Big(\frac{\rho_j - \theta}{\tau_{\mathrm{ce}}}\Big),$
>
> and first recover token-level gradients by scattering the upstream gradient of the compressed sequence back to the original positions (Eq. (11)). Using these gradients, Eq. (12) defines the Taylor-based importance $\eta_j = \langle (T^{\ell}\_{\mathrm{vis}})\_j, (\partial L / \partial T^\ell\_{\mathrm{vis}})\_j\rangle$, which measures the predicted loss change when token $ j $ is removed, while Eq. (13)-(14) map $ \eta_j $ to a bounded directional term $ d_j $ and Eq. (16) provides a positive scale $ s_j $. We then define the surrogate gradient on the relaxed retention probability as:
>
> $\frac{\partial L}{\partial q_j} \approx \beta\, d_j\, s_j\, \max\{q_j(1-q_j), \epsilon_{\text{sat}}\},$
>
> which is exactly what is implemented in our custom backward function. Because $q_j$ depends on $ \theta $ only through the sigmoid in Eq. (7), the chain rule in Appendix A.1.3 yields
>
> $\frac{\partial L}{\partial \theta} = \sum_j \frac{\partial L}{\partial q_j} \cdot \frac{\partial q_j}{\partial \theta}, \quad \frac{\partial L}{\partial \rho_j} = \frac{\partial L}{\partial q_j} \cdot \frac{\partial q_j}{\partial \rho_j},$
>
> so the threshold predictor is optimized only through the task loss and the Taylor-based importance signal, and once $L$ has approximately converged and the magnitudes of $\partial L/\partial q_j$ become small, the induced gradients on $ \theta $ also become negligible, causing the estimated thresholds and pruning rates to stabilize in practice. Theoretically, at any stationary point of this relaxed surrogate system, we have $\partial L/\partial q_j=0$ for all $j$. Given the strictly positive scaling factors, this implies $d_j=0$ and consequently, the standardized contributions $\eta_j$ are balanced under the learned mask. This yields pruning patterns that are first-order self-consistent with the loss geometry, while Appendix A.2.4 and Fig. 5-6 show empirically that the resulting dynamics converge to stable, non-degenerate retention bands in practice.
>
> **W3 Extend to more modalities.**
>
> Thank you for your feedback. As mentioned in the Limitation and Future Work section, while our experiments have primarily focused on CT data, we believe our method is extensible to other modalities, such as MRI. To validate this, we conducted experiments on MRI data from the SLAKE [7] dataset as an example:
>
> |SLAKE [7]|Average Token Keep Rate|Acc|Match/Total|
> |-|-|-|-|
> |Qwen2.5-VL 3B|1.000|0.8446|413/489|
> |Ours 3B|0.375|0.8425|412/489|
>
> The results demonstrate that, while achieving significant computational efficiency gains, our method suffers only minimal performance degradation. Furthermore, to further validate the applicability of our method to different tasks, we also tested it on the NExT-QA [8] dataset for natural video QA tasks:
>
> |NExT-QA [8]|Average Token Keep Rate|Causal-How|Causal-Why|Descriptive-Count|Descriptive-Location|Descriptive-Other|Temporal-Now|Temporal-Next|Temporal-Present|Average Acc|
> |-|-|-|-|-|-|-|-|-|-|-|
> |Qwen2.5-VL 3B|1.000|0.7545|0.7738|0.6429|0.9358|0.8533|0.7760|0.7269|0.7097|0.7729|
> |Ours 3B|0.473|0.7502|0.7693|0.6522|0.9358|0.8483|0.7631|0.7455|0.7527|0.7723|
>
> As a plug-and-play method, ITS and SGP can easily be extended to different modality tasks and achieved comparable gains.

---

> ### Author Response · Authors · 2025-11-22
> **Response to Reviewer TVyY (3/4)**
>
> **W4 Statistical analysis.**
>
> To account for randomness, we repeat the experiment of Table 4, with ten independent random seeds on DeepTumorVQA [9] under the same protocol and in medical metrics. We report mean $\pm$ standard deviation over these runs. Photon consistently outperforms the strongest baseline for all seeds, and the performance gains are much larger than the cross-seed standard deviation, indicating that the improvements are stable with respect to initialization.
>
> ||Sensitivity|Specificity|Accuracy|
> |-|-|-|-|
> |RadFM [10] 13B|75.2|98.6|88.6|
> |Merlin [11] 7B|50.2|51.2|50.7|
> |Ours 3B|81.2 $\pm$ 1.7|95.2 $\pm$ 0.1|89.3 $\pm$ 0.7|
>
> For the main Photon models, we also estimate test-set uncertainty via nonparametric bootstrap sampling 1000 times over evaluation questions, reporting 95 percent confidence intervals for sensitivity, specificity, and accuracy, these intervals are narrow, indicating that the reported gains are not explained by test-set sampling variability.
>
> ||Sensitivity|Specificity|Accuracy|
> |-|-|-|-|
> |RadFM [10] 13B|75.2|98.6|88.6|
> |Merlin [11] 7B|50.2|51.2|50.7|
> |Ours 3B|81.7 [78.6, 84.7]|95.3 [93.8, 96.6]|89.5 [87.9, 91.1]|
>
> **Q1 Open source.**
>
> Thank you for your comment. We agree that code availability is essential for reproducibility. We are currently organizing and documenting the full implementation, including the model code, training and evaluation scripts, preprocessing tools, and data loaders. After this process is completed, we will release the entire codebase to ensure full reproducibility and facilitate future research.
>
> To enhance reproducibility, all hyperparameters used in our experiments are listed in Appendix A.2.1, including the optimizer, learning rate schedule, training steps, gradient accumulation, weight decay, and fixed parameters related to SGP and token retention, ensuring that the core training setup can be fully reproduced.
>
> For datasets and preprocessing, Appendix A.2.2 specifies the exact training subsets used in Phase 1 and the patient counts, volume numbers, and QA instances for 3D-RAD and DeepTumorVQA in Phase 2, following the official evaluation protocols. We also adopt the official CT-RATE [12] preprocessing and resize volumes to keep the visual token count below 7K to avoid GPU memory overflow during baseline training. The complete data processing pipeline will be released with the code.
>
> For data loading, we provide a 3D volume loader based on SimpleITK for volume reading, sequence construction, and instruction concatenation, while preserving compatibility with the native Qwen2.5-VL data flow. This module will be included in the public repository so that researchers can construct training and evaluation inputs exactly as described.
>
> **Q2 Robustness in noisy scenario.**
>
> Thank you for your insightful question. We have evaluated the robustness of our method under noisy saliency estimates and threshold perturbations, and we would like to clarify the mechanisms in place to prevent model collapse.
>
> **Q2.1. Robustness to Noise in Saliency Estimates and Threshold Predictions:**
>
> For token score estimation, we use the attention scores produced by the model itself rather than introducing an additional estimation network. This keeps the pruning behavior consistent with the primary task model and avoids external interference. For threshold prediction, we evaluated robustness by injecting $ \pm $10% random noise into the estimated thresholds during inference. The results show that the performance drop under such non-destructive noise remains acceptable:
>
> ||Existence Detection|Static Temporal Diagnosis|Longitudinal Temporal Diagnosis|Medical Measurement|Image Observation|Anomaly Detection|
> |-|-|-|-|-|-|-|
> |Original Thresholds|83.07|52.86|77.01|37.74/39.36/96.14|51.59/56.66/93.62|42.33/47.50/91.96|
> |W/ threshold noise ([-10%,10%])|82.87|52.33|76.59|37.63/39.05/96.13|51.43/56.72/93.61|42.15/47.38/91.93|
>
> These results suggest that even with noisy threshold estimates, our method remains robust.

---

> ### Author Response · Authors · 2025-11-22
> **Response to Reviewer TVyY (4/4)**
>
> **Q2.2. Preventing Token Usage Degeneration During Training:**
>
> In contrast to prior methods which do not include explicit mechanisms to prevent collapse during training, our design accounts for the limited prior knowledge of the base model on 3D CT data. To avoid early removal of important tokens, we incorporate safeguards in both the training schedule and the design of SGP and regularization.
>
> As outlined in Appendix A.2.4 and shown in Figure 6, we use a staged training schedule:
>
> **Warmup-1:** No pruning, allowing the threshold predictor and visual-language features to converge on the full volume.
>
> **Warmup-2:** Soft masking is applied during the forward pass without hard pruning, allowing the thresholds and saliency to learn reasonable distributions under the condition where all tokens are retained.
>
> **Reduction Phase:** introduces hard pruning.
>
> This progression stabilizes threshold prediction before any token is removed and reduces the risk of losing critical information in early training.
>
> **SGP and regularization further prevent collapse.** The surrogate gradient updates thresholds based on the first-order loss change: if an important token is mistakenly pruned, the loss rises, and the gradient pushes its retention probability upward; redundant tokens receive the opposite update. The Soft Retention Band keeps the average retention rate away from extreme values and prevents "almost all pruned" or "almost all retained" states.
>
> Furthermore, our **Flip and Robustness Regularization** terms punish the model for ignoring visual tokens, encouraging the model to focus on visual information.
>
> The results shown in Figure 5, Figure 6, and Appendices A.2.3 and A.2.4 further validate the robustness and convergence of our method, confirming that no collapse occurs during token retention training.
>
> **Q3 Analysis on anatomically and clinically critical regions.**
>
> To address this, we performed a block level overlap analysis between retained tokens and reference lung segmentations on 3D volumes. We use pretrained segmentation model to get the reference mask. The reference masks were resampled to the token grid used by our pruning module. For each volume, we compute the fraction of positive lung blocks that are retained, and we also aggregate this over the full lung subset.
>
> The global and per-volume coverages in the table below show that the pruning module preserves most lung parenchyma blocks with consistent behavior across cases. Since some lesion-focused cases do not require observing the entire parenchyma, this results is resonable.
>
> ||Global coverage|Mean per volume coverage|
> |-|-|-|
> |Ours 3B|0.831|0.818|
>
>
> **Thank you for your valuable comment again, we hope our responses can address your concerns and we look forward to your further reply!**
>
> Reference:
>
> [1] ATP-LLaVA, CVPR 2025, Ye et al.
>
> [2] IVTP, ECCV2024, Huang et al.
>
> [3] TransPrune, arXiv 2025, Li et al.
>
> [4] VisionZip, CVPR 2025, Yang et al.
>
> [5] HiPrune, arXiv 2025, AAAI 2026 SAPP, Liu et al.
>
> [6] 3D-RAD, NeurIPS 2025, Gai et al.
>
> [7] SLAKE, ISBI 2021, Liu et al.
>
> [8] NExT-QA, CVPR 2021, Xiao et al.
>
> [9] DeepTumorVQA, arXiv 2025, Chen et al.
>
> [10] RadFM, Nature Communications 2025, Wu et al.
>
> [11] Merlin, Research Square 2024, Blankemeier et al.
>
> [12] CT-RATE, arXiv 2024, Hamamci et al.

---

### Official Review · Reviewer_6q9R · 2025-10-26

**Soundness:** 4
**Presentation:** 4
**Contribution:** 3
**Rating:** 6
**Confidence:** 5

**Summary:**

This paper proposes Photon, a 3D-native multimodal large language model (MLLM) for medical visual question answering (Med-VQA). Photon aims to overcome the computational burden of 3D medical imaging through two novel modules:

- Instruction-conditioned Token Scheduling (ITS): dynamically prunes visual tokens using instruction-aware saliency and an instance-specific threshold predictor.
- Surrogate Gradient Propagation (SGP): enables differentiable optimization despite discrete token dropping via surrogate gradient restoration.

Photon achieves substantial acceleration (up to 5× faster inference) and reduced GPU memory usage while maintaining or surpassing state-of-the-art accuracy on 3D-RAD and DeepTumorVQA benchmarks.

**Strengths:**

- The combination of *instruction-conditioned pruning* and *surrogate gradient backpropagation* for efficient 3D token handling is innovative and mathematically well-founded.


- Photon outperforms all major baselines (RadFM, M3D, OmniV, Lingshu, etc.) by **3–14%** across multiple Med-VQA tasks.
- Achieves ~2/3 GPU memory reduction and ~5× training/inference speedup, verified through detailed benchmarks and ablations.


- Includes both 3D-RAD and DeepTumorVQA, along with visualizations, ablation on ITS/SGP, and computational efficiency metrics.

**Weaknesses:**

- If the instruction-conditioned token scheduling can transfers to domains with different spatial and noise characteristics?

- How does Photon perform in zero-shot settings on out-of-distribution datasets?

- While comparisons are thorough within medical MLLMs, the study omits re-implementations of VisionZip, LLaVA-PruMerge, or ATP-LLaVA under medical conditions, limiting cross-domain efficiency comparisons.

- How would Photon scale when integrated with larger base models in terms of training stability and memory efficiency?

- Can the authors clarify if ITS + SGP could be extended to temporal or video-based reasoning, where token dynamics change over time?

- Certain figures lack descriptive captions or context. Even though the figures are well-designed and beautiful, I can not fully understand them without reading the main text carefully.

- Related works are missing some relevant medical vision language models, such as Med-R1 [1], HealthGPT [2].

[1] Lai, Yuxiang, et al. "Med-r1: Reinforcement learning for generalizable medical reasoning in vision-language models." arXiv preprint arXiv:2503.13939 (2025).

[2] Lin, Tianwei, et al. "Healthgpt: A medical large vision-language model for unifying comprehension and generation via heterogeneous knowledge adaptation." arXiv preprint arXiv:2502.09838 (2025).

**Questions:**

Please refer to the Weaknesses section.

**I am willing to raise my score according to the rebuttal.**

---

> ### Author Response · Authors · 2025-11-22
> **Response to Reviewer 6q9R (1/3)**
>
> We sincerely thank the reviewer for the time and valuable suggestions, and for recognizing the novelty of our instruction-conditioned pruning with surrogate gradients, Photon’s performance gains over major baselines, the memory and speed improvements, and the completeness of our evaluation across datasets and analyses.
>
> **W1 Transfer instruction-conditioned token scheduling to more domain with different spatial and noise characteristics.**
>
> Our Instruction-conditioned Token Scheduling (ITS) does not use fixed thresholds tied to raw intensities or spatial characteristics. It derives a task-dependent retention ratio from the instruction-visual cross-attention distribution, and its threshold is estimated using features that reflect both relative ordering and absolute scale. This design avoids completely relying on center-specific intensity ranges or noise patterns and remains adaptable across different imaging centers, protocols, and noise levels.
>
> Our results reveal that the same ITS configuration yields stable gains on datasets with large differences in spatial range, task types, and annotation styles, including 3D-RAD [1] and DeepTumorVQA [2], showing that ITS captures a task-guided token-importance pattern.
>
> We further validated this on MedFrameQA [3], which includes diverse modalities with varying frame counts, slice numbers, orientations, spacing, and resolutions from multiple sources. Using a random 9:1 train-test split, our method matches the performance of the non-pruned model, confirming robustness under varying noise and spatial characteristics.
>
> |MedFrameQA [3]|Average Token Keep Rate|CT|MRI|X-ray|Other|Ultrasound|Overall|
> |-|-|-|-|-|-|-|-|
> |Qwen2.5- VL 3B|1.00|0.744|0.745|0.893|0.667|0.486|0.727|
> |Ours 3B|0.432|0.761|0.787|0.857|0.750|0.486|0.745|
>
> **W2 Zero-shot setting on out-of-domain datasets.**
>
> Thank you for your suggestion. We first used the AbdomenAtlas3.0 OOD split (sharing the same data source as DeepTumorVQA) to divide DeepTumorVQA into training and testing sets. This split separates institutions, resulting in differences in population, scanners, and imaging parameters. The multi-choice and free-text results are shown below.
>
> |DeepTumorVQA OOD|Measurement|Recognition|Visual Reasoning|Medical Reasoning|Overall|AVG Token|
> |-|-|-|-|-|-|-|
> |Multi-Choice|0.745|0.599|0.636|0.677|0.648|0.34K|
> |Free-Text|0.459|0.643|0.520|0.599|0.570|0.40K|
>
> Compared with the in-domain results in Table 2, the OOD scores drop only slightly, by 0.038 for multi-choice (0.686 in-domain) and 0.049 for free-text (0.619 in-domain), showing that our method preserves strong performance under OOD shifts while providing efficiency gains.
>
> To further evaluate our pruning methods influence on resutls in a zero-shot setting, we tested Photon without any additional medical data training on MedFrameQA. The results are shown below.
>
> |MedFrameQA [3]|Average Token Keep Rate|CT|MRI|X-ray|Other|Ultrasound|Overall|
> |-|-|-|-|-|-|-|-|
> |Qwen2.5- VL 3B Zero-Shot|7.00K|0.461|0.460|0.485|0.477|0.432|0.460|
> |Qwen2.5-VL 3B + ITS 3B Zero-Shot|3.28K|0.459|0.456|0.488|0.470|0.426|0.457|
>
> The token scheduling strategy in Photon is isolated from the MLLM backbone and operates as a plugin, meaning it does not affect the backbone's zero-shot or generalization ability. Experimental results confirm that our approach remains effective in zero-shot setting.

---

> > ### Author Response · Authors · 2025-11-22
> > **Response to Reviewer 6q9R (2/3)**
> >
> > **W3 Compare with other token pruning methods.**
> >
> > Thank you for your valuable feedback. Because ATP-LLaVA [4] has not been open-sourced, and LLaVA-PruMerge [5] is not suitable for Qwen, we discussed them in related works and we added two baselines: VisionZip [6] and HiPrune [7], adapted for 3D medical tasks, ensuring a fair experimental setup.
> >
> > Because prior methods do not support instruction-specific adaptive retention, we evaluate them at fixed 30%, 50%, and 70% retention levels. These methods were not designed for training-time acceleration or stability, and many do not provide training procedures. In 3D medical settings, where the base model has limited prior knowledge, applying them during training can remove essential tokens too early and cause language-only degeneration. To ensure fairness, we follow their recommended usage: we train the full Qwen2.5-VL model without pruning and evaluate their pruning strategies with the official code based solely on prediction performance.
> >
> > |Methods(Keep Vision Token Per Sample)|Existence Detection|Static Temporal Diagnosis|Longitudinal Temporal Diagnosis|Medical Measurement|Image Observation|Anomaly Detection|Infer Speed|
> > |-|-|-|-|-|-|-|-|
> > |Qwen2.5 VL (7.0K)|81.97|47.62|75.36|37.00/38.57/96.04|52.72/56.48/93.63|42.01/47.36/91.93|2.30 Tok/s|
> > |VisionZip (2.1K)|82.00|47.19|75.42|37.04/38.69/96.06|52.51/56.36/93.63|41.96/47.39/91.93|2.32 Tok/s|
> > |VisionZip (3.5K)|81.99|47.74|75.93|37.11/38.68/96.07|52.72/56.60/93.65|41.98/47.36/91.92|2.23 Tok/s|
> > |VisionZip (4.9K)|82.01|47.40|75.88|37.07/38.82/96.04|52.57/56.43/93.64|42.01/47.39/91.93|2.18 Tok/s|
> > |HiPrune (2.1K)|81.99|48.08|75.50|37.00/38.59/96.05|52.56/56.46/93.64|42.08/47.32/91.93|0.76 Tok/s|
> > |HiPrune (3.5K)|81.97|47.84|75.58|37.14/38.75/96.07|52.65/56.60/93.63|41.92/47.31/91.91|0.75 Tok/s|
> > |HiPrune (4.9K)|82.02|47.19|75.35|37.00/38.62/96.06|52.57/56.44/93.62|41.94/47.40/91.91|0.73 Tok/s|
> > |Photon (Dynamic)|83.07|52.86|77.01|37.74/39.36/96.14|51.59/56.66/93.62|42.33/47.50/91.96|4.12 Tok/s|
> > |Photon MAX(Dynamic)|85.50|57.79|79.70|39.44/40.96/96.17|52.83/58.72/93.70|42.85/48.89/92.00|2.60 Tok/s|
> >
> >
> > Most previous pruning methods, including VisionZip [6] and HiPrune [7], are either incompatible with Flash Attention in the visual encoder or require extra attention computation across all encoder layers, which reduces efficiency. In our experiments, each case has many visual tokens with short answer, under this scenario, these methods offer no speed advantage and degrade more accuracy. Our method provides both higher accuracy and faster inference.
> >
> >
> >
> > **W4 Evaluating with larger model.**
> >
> > Thank you for your question. To further demonstrate the scalability of our method, we extended our evaluation to a 7B model. As shown in the table below, the 7B model outperforms the 3B model, confirming that our approach scales effectively with larger model sizes. Additionally, the average token retention rate shows that the larger model still maintains token efficiency.
> >
> > ||Existence Detection|Static Temporal Diagnosis|Longitudinal Temporal Diagnosis|Medical Measurement|Image Observation|Anomaly Detection|Average Keep Token Num|
> > |-|-|-|-|-|-|-|-|
> > |Ours 3B|83.07|52.86|77.01|37.74/39.36/96.14|51.59/56.66/93.62|42.33/47.50/91.96|0.39K|
> > |Ours 7B|83.50|53.90|78.03|40.70/41.90/96.20|53.34/57.65/93.77|42.66/48.14/91.99|0.41K|

---

> ### Author Response · Authors · 2025-11-22
> **Response to Reviewer 6q9R (3/3)**
>
> **W5 Extend to video based reasoning.**
>
> Thank you for your question. As a plug-and-play method, both ITS and SGP can be easily extended to video understanding tasks, potentially with simpler training schemes than in 3D medical tasks, while achieving similar performance benefits. In the context of video tasks, since the base model has already been well pre-trained on visual data, the two warmup stages and our designed robustness regularization can be removed.
>
> To demonstrate the applicability of ITS and SGP for video reasoning, we evaluated their performance on the NExT-QA [8] video question-answer dataset, with the results shown below:
>
> |NExT-QA|Average Token Keep Rate|Causal-How|Causal-Why|Descriptive-Count|Descriptive-Location|Descriptive-Other|Temporal-Now|Temporal-Next|Temporal-Present|Average Acc|
> |-|-|-|-|-|-|-|-|-|-|-|
> |Qwen2.5-VL 3B|1.000|0.7545|0.7738|0.6429|0.9358|0.8533|0.7760|0.7269|0.7097|0.7729|
> |Ours 3B|0.473|0.7502|0.7693|0.6522|0.9358|0.8483|0.7631|0.7455|0.7527|0.7723|
>
>
> These results show that ITS and SGP can be successfully applied to video tasks, achieving comparable performance while significantly reducing the number of retained tokens.
>
>
>
>
>
> **W6 Description of figures.**
>
> Thank you for your valuable comment. We have revised the figures so that each one can be understood without relying on the main text. We refined all captions to directly describe the visual elements and results, and we redrew the key illustrations to improve clarity.
>
> Additionally, In Figure 1, we remade the bar chart and radar plot with a clearer mapping between methods and results, and we redesigned the upper-right illustration to better show the working modes of ITS and SGP. In Figure 2, we added module boundaries, applied a clearer color scheme for frozen components, and updated the caption accordingly. We also added separating lines and clearer labels across groups of images in Figure 5. These changes make the figures more self-contained and easier to interpret at a glance.
>
>
>
> **W7 Discussing more related works.**
>
> Thank you for the suggestion. In the revised version of related works, we have added both Med-R1 [9], which applies reinforcement learning and chain-of-thought reasoning to medical tasks, and HealthGPT [10], which integrates image generation and understanding within a unified medical multimodal framework. We also discussed how these works relate to our setting, which strengthens the background section.
>
>
>
> **Thank you for your valuable comment again, we hope our responses can address your concern and we look forward to your further reply!**
>
> Reference:
>
> [1] 3D-RAD, NeurIPS 2025, Gai et al.
>
> [2] DeepTumorVQA, arXiv 2025, Chen et al.
>
> [3] MedFrameQA, arXiv 2025, Yu et al.
>
> [4] ATP-LLaVA, CVPR 2025, Ye et al.
>
> [5] LLaVA-PruMerge, ICCV 2025, Shang et al
>
> [6] VisionZip, CVPR 2025, Yang et al.
>
> [7] HiPrune, arXiv 2025, AAAI 2026 SAPP, Liu et al.
>
> [8] NExT-QA, CVPR 2021, Xiao et al.
>
> [9] Med-R1, arXiv 2025, Lai et al.
>
> [10] HealthGPT, ICML 2025, Lin et al.

---

### Official Review · Reviewer_goC1 · 2025-10-30

**Soundness:** 3
**Presentation:** 2
**Contribution:** 2
**Rating:** 4
**Confidence:** 3

**Summary:**

This paper proposes a 3D-native framework for applying MLLM to 3D medical volumes. Unlike existing methods that mainly take 2D slices as input, their method instead directly works with 3D volumes using variable-length visual token sequences. Core to this method are 1) Instruction-conditioned Token Scheduling (ITS) which dynamically prunes visual tokens based on the text instruction and 2) Surrogate Gradient Propagation (SGP), a custom gradient estimation technique to enable end-to-end training of the discrete token-dropping mechanism. Experiments on several benchmarks show that this method improves performance while being efficient.

**Strengths:**

1. This work addresses an important bottleneck in medical AI (computational challenge for medical QA with MLLM due to 3D nature of the data). The motivation is clear, as processing full 3D volumes preserves volumetric information while dynamic token selection is relatively computationally affordable.
2. The proposed method IST is novel and seems to be effective. Instruction-conditioned, instance-adaptive token pruning is a step beyond common, instruction-agnostic pruning or fixed-ratio compression methods.
3. Ablations and computation footprint are transparent (tab 3)

**Weaknesses:**

1. The proposed method is overly complex, making it hard to follow let alone reproduce. For example, the derivation of the final surrogate gradient involves a long chain of heuristic-based calculations like standardization z_j, monotonic mapping r_j, directional term d_j, magnitude term m_j, and several clipping and clamping operations. Then there are also several regularization terms. These combined make the proposed method inherently brittle. This high sensitivity to implementation details not only hinders reproducibility but also questions the robustness of the reported performance and efficiency gains
2. Missing comparison to other pruning baselines. The paper mentions other pruning methods like LLaVA-PruMerge and ATP-LLaVA, but did not adapt them as a comparison. The authors mention that these methods lack capabilities such as instruction awareness, etc., but without direct comparison, there is no evidence for such claims. I believe the authors should have adapted at least one other baseline to the 3D medical domain as a more direct baseline to demonstrate the superiority of the proposed ITS and SGP mechanisms, especially since the authors sell on efficiency.
3. Is there any coefficient for the regularization terms? How sensitive is the method to each of them? I think more explanatioon is needed here.

**Questions:**

Please see the weaknesses.

---

> ### Author Response · Authors · 2025-11-22
> **Response to Reviewer goC1 (1/2)**
>
> We sincerely thank the reviewer for the time and valuable suggestions, and for recognizing the importance of addressing the 3D computational bottleneck in medical QA, the novelty and effectiveness of our instruction-conditioned adaptive pruning, and the clarity of our ablations and computational analysis.
>
> **W1 Analysis of reproducibility.**
>
> The **normalization operations** are introduced to keep the operations within a manageable range for the subsequent steps. For soft retention with **regularization**, removing it will not cause the method to fail. As shown in Figure 5, this regularization is rarely triggered and is mainly intended to avoid extreme situations that could destabilize training.
>
> 1) **Our method** relies on a stable, differentiable design based on straight-through estimator and first-order Taylor importance estimation, rather than heuristic rules. The gradient used for threshold learning in SGP comes from a first-order estimate of loss change when dropping a visual token, consistent with the criterion of Molchanov et al. (Appendix A.1.4), which ensures stability and effectiveness.
>
> 2) Although **variables** such as $z_j$, $r_j$, $d_j$, and $m_j$ in ITS and SGP add complexity to the notation, we include them explicitly to ensure transparency. Ablations show that removing ITS or SGP reduces performance. Photon achieves equal or better accuracy with faster training even when retaining fewer tokens.
>
> 3）Regarding **hyperparameters**, it is worth noting that not all hyperparameters are proposed by our method. We provided detailed descriptions of all parameters involved for better reproducibility. We further ablated the method-specific hyperparameters on 3D-RAD (table below). The results on 3D-RAD indicate that different settings mainly affect the average number of retained tokens, without significantly reducing task accuracy.
>
> ||Value|Existence Detection|Static Temporal Diagnosis|Longitudinal Temporal Diagnosis|Medical Measurement|Image Observation|Anomaly Detection|Average Keep Token Num|
> |-|-|-|-|-|-|-|-|-|
> |||ACC|ACC|ACC|BLEU/ROUGE/BERTScore|BLEU/ROUGE/BERTScore|BLEU/ROUGE/BERTScore||
> |$ \tau_{\text{ce}} $|0.10|82.44|51.66|77.66|38.65/39.87/96.20|51.11/56.77/93.61|41.95/46.91/91.86|0.42K|
> ||0.20|83.07|52.86|77.01|37.74/39.36/96.14|51.59/56.66/93.62|42.33/47.50/91.96|0.39K|
> ||0.50|82.88|52.13|77.71|38.59/39.77/96.25|51.61/56.16/93.53|41.73/46.81/91.87|0.32K|
> |$ \beta $|0.5|82.99|52.25|75.57|37.05/38.45/96.04|51.60/56.12/93.64|41.76/47.18/91.90|0.46K|
> ||0.8|83.07|52.86|77.01|37.74/39.36/96.14|51.59/56.66/93.62|42.33/47.50/91.96|0.39K|
> ||1.0|82.70|52.86|76.63|38.51/39.70/96.11|52.08/56.79/93.72|41.99/47.52/91.93|0.40K|
> |$ \epsilon_{\text{sat}} $|1e-3|82.87|51.53|76.78|37.09/38.29/96.00|52.14/56.72/93.66|42.20/47.22/91.97|0.45K|
> ||5e-3|83.07|52.86|77.01|37.74/39.36/96.14|51.59/56.66/93.62|42.33/47.50/91.96|0.39K|
> ||1e-2|82.83|53.27|76.40|37.67/39.10/96.20|51.27/56.11/93.65|41.78/47.08/91.91|0.61K|
>
> Additionally, we have applied the same hyperparameter configuration across a broader set of datasets, including NExT-QA [3] for general video tasks, SLAKE [4] for MRI VQA, and MedFrameQA [5] for multi-frame, multi-modal medical VQA.
>
> For MedFrameQA, we randomly split the dataset into training and testing sets (9:1 ratio).
>
> |MedFrameQA [5]|Average Token Keep Rate|CT|MRI|X-ray|Other|Ultrasound|Overall|
> |-|-|-|-|-|-|-|-|
> |Qwen2.5-VL 3B|1.000|0.744|0.745|0.893|0.667|0.486|0.727|
> |Ours 3B|0.432|0.761|0.787|0.857|0.750|0.486|0.745|
>
>
> For NExT-QA, we use the official dataset split to train and test.
>
> |NExT-QA [3]|Average Token Keep Rate|Causal-How|Causal-Why|Descriptive-Count|Descriptive-Location|Descriptive-Other|Temporal-Now|Temporal-Next|Temporal-Present|Average Acc|
> |-|-|-|-|-|-|-|-|-|-|-|
> |Qwen2.5-VL 3B|1.000|0.7545|0.7738|0.6429|0.9358|0.8533|0.7760|0.7269|0.7097|0.7729|
> |Ours 3B|0.473|0.7502|0.7693|0.6522|0.9358|0.8483|0.7631|0.7455|0.7527|0.7723|
>
>
> For SLAKE dataset, we extract the MRI subset and use official dataset split to train and test.
>
> |SLAKE [4]|Average Token Keep Rate|ACC|Match Case/Total Case|
> |-|-|-|-|
> |Qwen2.5-VL 3B|1.000|0.8446|413/489|
> |Ours 3B|0.375|0.8425|412/489|
>
>
> The experimental results demonstrate that, while our method offers a variety of hyperparameters for tuning, the current settings yield good performance in most cases, highlighting that our method is not a fragile implementation.

---

> ### Author Response · Authors · 2025-11-22
> **Response to Reviewer goC1 (2/2)**
>
> **W2 Compare with other token pruning methods.**
>
> Thank you for your valuable feedback. We have now added two direct baselines: **VisionZip [6]** and **HiPrune [7]**, adapted for 3D medical tasks, ensuring fair experimental settings.
>
> Since prior methods do not support instruction-dependent adaptive retention, we tested them at fixed 30%, 50%, and 70% retention ratios. These methods were not designed for training-time acceleration or stability, and many do not provide training procedures. In the 3D medical setting, where the base model has limited prior knowledge, directly training them can remove essential tokens too early and collapse into language bias. Therefore, we followed their recommended paradigms, trained the full Qwen2.5-VL model without pruning to ensure proper optimization, and evaluated their pruning strategies using the official code, focusing solely on prediction performance for fairness.
>
> |Methods (Keep Vision Token Per Sample)|Existence Detection|Static Temporal Diagnosis|Longitudinal Temporal Diagnosis|Medical Measurement|Image Observation|Anomaly Detection|Infer Speed|
> |-|-|-|-|-|-|-|-|
> |Qwen2.5-VL (7.0K)|81.97|47.62|75.36|37.00/38.57/96.04|52.72/56.48/93.63|42.01/47.36/91.93|2.30 Tok/s|
> |VisionZip (2.1K)|82.00|47.19|75.42|37.04/38.69/96.06|52.51/56.36/93.63|41.96/47.39/91.93|2.32 Tok/s|
> |VisionZip (3.5K)|81.99|47.74|75.93|37.11/38.68/96.07|52.72/56.60/93.65|41.98/47.36/91.92|2.23 Tok/s|
> |VisionZip (4.9K)|82.01|47.40|75.88|37.07/38.82/96.04|52.57/56.43/93.64|42.01/47.39/91.93|2.18 Tok/s|
> |HiPrune (2.1K)|81.99|48.08|75.50|37.00/38.59/96.05|52.56/56.46/93.64|42.08/47.32/91.93|0.76 Tok/s|
> |HiPrune (3.5K)|81.97|47.84|75.58|37.14/38.75/96.07|52.65/56.60/93.63|41.92/47.31/91.91|0.75 Tok/s|
> |HiPrune (4.9K)|82.02|47.19|75.35|37.00/38.62/96.06|52.57/56.44/93.62|41.94/47.40/91.91|0.73 Tok/s|
> |Photon (Dynamic)|83.07|52.86|77.01|37.74/39.36/96.14|51.59/56.66/93.62|42.33/47.50/91.96|4.12 Tok/s|
> |Photon MAX (Dynamic)|85.50|57.79|79.70|39.44/40.96/96.17|52.83/58.72/93.70|42.85/48.89/92.00|2.60 Tok/s|
>
>
> Most previous pruning methods, including VisionZip [6] and HiPrune [7], are not optimized for compatibility with Flash Attention in the visual encoder or require extra attention computation across all encoder layers, leading to notable performance loss. Consequently, when many visual tokens and short answers are involved, these methods do not provide clear prediction-speed benefits in our setting, they even slow down inference and yield suboptimal accuracy. In contrast, our method achieves both higher accuracy and faster inference.
>
>
>
> **W3 Analysis of regularization.**
>
> Thank you for your question. We would like to clarify that the flip and robustness regularization do not run simultaneously with the task loss (CE Loss) in the same batch. In regularization batches, we replace the task loss with the corresponding regularization loss, so no additional coefficients are required. During the forward pass, flip regularization is used when retained tokens are flipped, robustness regularization is used during data augmentation, and the task loss is used when no extra operation is applied. This setup encourages the model to rely on visual evidence without adding auxiliary tasks (such as OCR) or inducing language bias. We have revised the description of the paper to avoid ambiguity.
>
> As for the soft retention with regularization, its value range is controllable and is triggered only when unreasonable token retention rates occur (such as near full retention or excessive pruning). It does not conflict with the task loss, so it does not require a dedicated coefficient.
>
> It is worth noting that all regularizers are optional. On tasks where the model has sufficient prior knowledge, we can remove these regularizers and still carry out stable training (e.g., the experiments we provided on NExT-QA [3] dataset in W1).
>
>
>
> **Thank you for your valuable comment again, we hope our responses can address your concern and we look forward to your further reply!**
>
> Reference:
>
> [1] 3D-RAD, NeurIPS 2025, Gai et al.
>
> [2] DeepTumorVQA, arXiv 2025, Chen et al.
>
> [3] NExT-QA, CVPR 2021, Xiao et al.
>
> [4] SLAKE, ISBI 2021, Liu et al.
>
> [5] MedFrameQA, arXiv 2025, Yu et al.
>
> [6] VisionZip, CVPR 2025, Yang et al.
>
> [7] HiPrune, arXiv 2025, AAAI 2026 SAPP, Liu et al.

---

### Official Review · Reviewer_buqX · 2025-10-30

**Soundness:** 4
**Presentation:** 2
**Contribution:** 4
**Rating:** 8
**Confidence:** 3

**Summary:**

The paper proposed Photon, a novel MLLM for efficient 3D medical volume VQA. It introduces the Instruction-conditioned Token Scheduling (ITS) module for adaptive visual token pruning based on its correspondence with the instruction and the Surrogate Gradient Propagation (SGP) module to enable end-to-end training while doing discrete token-dropping. The proposed method can efficiently reduce the computational cost during both training and inference, and it also improves the performance on 2 public 3D medical VQA datasets with a nontrivial improvement.

**Strengths:**

1. The proposed method shows a significant efficiency improvement without sacrificing its performance, which is quite impressive. According to the experiments, the proposed method can successfully reduce the token length by more than 50%, speed up the training process by over 5 times, and reduce the memory usage at the same time. It also shows a non-trivial improvement against SoTA baselines, including larger ones, like the 7B Lingshu model.

2. The idea of instruction-conditioned token scheduling is quite novel to the reviewer. The ITS module makes the token-dropping process instruction conditioned by measuring the saliency against the instruction tokens, providing a dynamic dropping based on different tasks. It also uses an adaptive threshold based on a series of crafted features to avoid over- or under-pruning.

3. The ablation experiment is very detailed, which is helpful to understand the contribution of different modules/losses/regularizers. The visualization in Figure 5 shows the necessity of the proposed regularizations, provides strong support to the model design.

**Weaknesses:**

The reviewer has 2 major concerns about this paper.

1. The paper itself is well-written, although it is not that easy to fully understand all the equations; it is at least clear and detailed. However, the figures in the paper, especially Figure 1 and Figure 2, are very difficult to follow. The reviewer understands that using a uniform color scheme can make it look nicer, but it should not harm the readability. For example, (a) The performance bar plot and radar plot at the bottom of Figure 1 are very confusing; the chosen color is so similar that it makes it difficult to tell the difference. (b) The upper right part of Figure 1 is also confusing; the idea of ITS and SGP is not shown here. (c) In the upper right part of Figure 2, the author uses 2 different colors for each module, which I guess refers to training/frozen, but it is still hard to tell, and there is no annotation for this. (d) The white mask in all qualitative results is difficult to distinguish, as well. It would be better if using some other way to highlight the retained patches, e.g., highlight the contour. (e) Figure 5, maybe better to add 2 separate lines between each figure, only using the color still makes it hard to tell which is which. Also, there are some small typos in the paper, like in equation (3), the bottom part of the summation notation should just be $t'\in\mathcal{Q}$, as $t$ is indicated by $c_t$ in the left-hand side.

2. While the proposed method shows nice results, it also increases the complexity of the framework by a lot. There are a lot more hyperparameters introduced here, and there is no explanation or guidance on how to select them. The SGP module itself contributes 6 new hyperparameters, and there are more in the whole framework, like where to insert the ITS and SGP module, and so on. It would be much better if the authors could provide some insight or analysis on how those hyperparameters are chosen and how sensitive the proposed method is to these hyperparameters.

3. While the paper already includes tons of technical details, it still lacks some high-level intuition on its design. Some of the design is not that intuitive to the reviewer. This makes it hard to fully understand the design. Please see the questions below.

4. While the paper already provided some visualization and failure case analysis, it would be better to include some more in-depth analysis on the reasons for failure. Also, the reviewer would like to see an analysis of the **instruction-conditioned** part of the token-dropping. Namely, is there a statistical difference in the token-dropping given different instructions? Given a different task, a different region of interest, will the model behave differently and choose to drop different tokens? This is very critical to support the paper's major claim.

**Questions:**

1. What is the high-level idea to use the text token affinity as the weight for the visual saliency score? The paper claims that tokens with higher overall affinity/attention score will have a higher weight, but what is the insight behind this? Intuitively, you may want to assign a higher weight to task-related word tokens like action/organ/location, and so on. But how does this relate to the text token affinity score?

2. Equation (6) uses 3 feature vectors to construct the final feature for the probability threshold. Can you provide some high-level ideas on this?

3. In the Flip regularizer, the model penalizes the model for giving high-confidence results when tokens that should be dropped are kept. This is a little bit counterintuitive, since it keeps the tokens that should be dropped is technically equivalent to adding unnecessary information. The needed tokens are still the same; the module can make a confident answer with some extra unnecessary information. A more intuitive case may be to reverse the tokens that should be kept, ie, remove necessary information. In this case, the model should give a low-confidence or even wrong answer. The reviewer is curious about this; some clarification would be great.

4. Figure 1 shows the training speed using a 32B level model, but it is not presented in the result section. Considering that the paper is only evaluating over 3 B-level models, it would be great if there were more results on larger models, such as 7B or 32B. Demonstrating the scaling capability.

5. The proposed method is only evaluated on Qwen2.5-VL, but it would be interesting to see its behavior on some other architectures, like LLaMA or Gemma. But the reviewer understands this will take a long time and a lot of work, so this would not influence my final decision. Just curious if authors have tried it on different MLLMs.

---

> ### Author Response · Authors · 2025-11-22
> **Response to Reviewer buqX (1/3)**
>
> We sincerely thank the reviewer for the time and valuable suggestions, and for recognizing the efficiency gains of our method, its improvements over strong baselines, the novelty of instruction-conditioned token scheduling with adaptive thresholds, and the clarity of our ablations and visualizations.
>
> **W1: Paper writing and figure drawing.**
>
> Thank you for the detailed feedback. We have addressed each point in the revised version:
> **(a)** We redrew the performance bar plot and radar plot in Figure 1. The bar plot now aligns each method with its values and uses clearer color separation to highlight our method. The radar plot adopts higher-contrast colors to distinguish the three methods. **(b)** We redesigned the upper-right illustration in Figure 1 to present the working modes of ITS and SGP more clearly. **(c)** In Figure 2, we added module boundaries for better contrast and used distinct colors for frozen modules, which is now explained in the caption. **(d)** In the qualitative visualizations, we kept the purple bounding boxes and descriptions for retained regions and reduced the opacity of the white dropped regions to make the separation between retained and dropped areas clearer. **(e)** In Figure 5, we added separating lines between illustration groups, refined the labels. We corrected the typographical errors in equations.
>
> Thank you again for your suggestions. We will continue improving readability and visual clarity in subsequent versions.
>
>
> **W2: Selection and analysis of hyperparameters.**
>
> Thank you for the comment. The hyperparameters we introduced mainly serve to stabilize training and prevent extreme behaviors, and we use one default configuration across all tasks in the paper. SGP-related hyperparameters only influence training, not inference. Below, we explain their roles and when they should be adjusted.
>
> **W2.1 Stability-oriented hyperparameters (no frequent tuning required):**
>
> + The normalization constant $\epsilon_{\text{std}}$ prevents instability due to small standard deviations, generally not adjusted.
> + The retention-ratio bounds $(r_{\min},\, r_{\max})$ prevent collapse toward keeping nearly all tokens or pruning excessively. As shown in Figure 5, this penalty is almost never activated once training stabilizes, and can be removed when the base model and data quality are sufficiently strong.
> + The scaling limits $s_{\min}$ and $s_{\max}$ control relative gradient magnitudes across tokens, constraining relative gradient differences.
>
> **W2.2 Pruning-related hyperparameters (direct impact on pruning behavior):**
>
> + $\tau_{\text{ce}}$ controls the smoothness of $q_j$, it was set to 0.2 in our experiments. When training is unstable or shaking, slightly increase to make it smoother, slightly reduce when stronger binarization is needed.
> + The surrogate gradient weight $\beta$ balances SGP with the main task loss, it was set to 0.8 in our experiments. Decrease if pruning oscillates too much, increase if convergence speed is slow.
> + The saturation threshold $\epsilon_{\text{sat}}$ prevents vanishing gradients when $ q_j $ is near 0 or 1, it was set to $5 \times 10^{-3}$ in our experiments, it should be increased when gradients near extreme values become too weak and should be decreased when the gradients around these extreme values become overly amplified.
>
> **W2.3 Placement of ITS/SGP.**
>
> + We insert ITS/SGP at $\ell=\ell_{n/4}$. The comparison in Table 3 shows that our setting and $\ell=\ell_{n/2}$ yield similar accuracy under matched retention levels, with our setting offering a slightly better tradeoff on accuracy and computational cost.
>
> We further conducted ablations on the above three hyperparameters that most directly affect retention. The results on 3D-RAD [1] show that different settings primarily change the average number of retained tokens rather than significantly degrading task accuracy.
>
> | | |Existence Detection |Static Temporal Diagnosis |Longitudinal Temporal Diagnosis |Medical Measurement |Image Observation |Anomaly Detection |Average Keep Token Num |
> |-|-|-|-|-|-|-|-|-|
> ||Value|ACC|ACC|ACC|BLEU/ROUGE/BERTScore|BLEU/ROUGE/BERTScore|BLEU/ROUGE/BERTScore||
> |$ \tau_{\text{ce}} $|0.10|82.44|51.66|77.66|38.65/39.87/96.20|51.11/56.77/93.61|41.95/46.91/91.86|0.42K|
> ||0.20|83.07|52.86|77.01|37.74/39.36/96.14|51.59/56.66/93.62|42.33/47.50/91.96|0.39K|
> ||0.50|82.88|52.13|77.71|38.59/39.77/96.25|51.61/56.16/93.53|41.73/46.81/91.87|0.32K|
> |$ \beta $|0.5|82.99|52.25|75.57|37.05/38.45/96.04|51.60/56.12/93.64|41.76/47.18/91.90|0.46K|
> ||0.8|83.07|52.86|77.01|37.74/39.36/96.14|51.59/56.66/93.62|42.33/47.50/91.96|0.39K|
> ||1.0|82.70|52.86|76.63|38.51/39.70/96.11|52.08/56.79/93.72|41.99/47.52/91.93|0.40K|
> |$ \epsilon_{\text{sat}} $|1e-3|82.87|51.53|76.78|37.09/38.29/96.00|52.14/56.72/93.66|42.20/47.22/91.97|0.45K|
> ||5e-3|83.07|52.86|77.01|37.74/39.36/96.14|51.59/56.66/93.62|42.33/47.50/91.96|0.39K|
> ||1e-2|82.83|53.27|76.40|37.67/39.10/96.20|51.27/56.11/93.65|41.78/47.08/91.91|0.61K|

---

> ### Author Response · Authors · 2025-11-22
> **Response to Reviewer buqX (2/3)**
>
> (Continuing from above) Adjusting these values is typically unnecessary unless instability occurs. The current settings are robust across various datasets and tasks, including 2D and natural-image video extensions (as shown in the revised appendix). Here is an example about NExT-QA [2] benchmark which is generally used for general-video question-answering tasks:
>
> |NExT-QA [2]|Average Token Keep Rate|Causal-How|Causal-Why|Descriptive-Count|Descriptive-Location|Descriptive-Other|Temporal-Now|Temporal-Next|Temporal-Present|Average Acc|
> |-|-|-|-|-|-|-|-|-|-|-|
> |Qwen2.5-VL 3B|1.000|0.7545|0.7738|0.6429|0.9358|0.8533|0.7760|0.7269|0.7097|0.7729|
> |Ours 3B|0.473|0.7502|0.7693|0.6522|0.9358|0.8483|0.7631|0.7455|0.7527|0.7723|
>
> The consistent performance confirms that the selected hyperparameters are robust.
>
> **W3: High-level description.**
>
> **Below, we provide high-level explanations to address your concerns.**
>
> **Q1: Intuition Behind Text Token Affinity for Visual Saliency**
>
> Text token affinity is used as the weight for visual saliency because self-attention reflects the semantic importance learned from the corpus. Tokens such as "nodule", "lobe", or "pneumonia", which are closely tied to diagnostic outcomes and frequently appear in the training data, receive more attention from other tokens. This centrality arises from the model’s attention distribution and encodes their structural role in task semantics. Using these affinities ensures that the model assigns higher saliency to the visual regions associated with these task-critical tokens.
>
> **Q2: The 3 feature vectors used in eq.6.**
>
> We use three feature vectors in Equation (6) to estimate the pruning threshold $ \theta $ because the number of tokens can vary across samples, making it difficult to use the full score sequence directly. With varying token lengths, padding or truncating would be required, which either leads to inefficiencies or results in the loss of important information. Processing the entire sequence requires handling more tokens than necessary, which is counterproductive to the goal of pruning. By using three statistical features,$ \Psi(\rho) $ (distribution parameters from normalized scores to captrue relative scale), $ \Phi(u) $ (distribution parameters from raw scores to capture absolute scale), and $ \Upsilon(u) $ (the mean and maximum values of the log-transformed scores), we capture the essential information needed for threshold estimation without needing to process the full sequence.
>
> **Q3: The design of Flip regularizer.**
>
> Actually, the Flip Regularizer not only keep the unnecessary token, but also removes the necessary token. If the model predicts the ground truth with **high confidence** given only the flipped mask, it implies reliance on text priors (hallucination). The regularizer penalizes this unjustified confidence, forcing the model to be uncertain when visual evidence is missing and grounding predictions in the retained features.
>
> **W4.1: Failure case analysis and instruction-conditioned token pruning analysis.**
>
> Regarding failure cases, we group errors into two types. **The first type** concerns small lesion measurements, which are harder than existence detection due to limited annotations, non-standardized labeling, and the need to map continuous values to discrete words. Although all methods perform worse on this subtask, Photon still matches or surpasses non-pruned baselines, indicating that these errors stem from task and supervision constraints rather than pruning instability.
>
> **The second type** involves rare or anatomically atypical cases. Long-tail distributions in training cause the model to rely more on common structures and exhibit higher uncertainty in uncommon regions. For instance, in 3D-RAD’s existence detection task, the positive sample counts for several representative findings lying in the tail of the distribution are:
>
> |Total case of these three|Bronchiectasis positive case|Peribronchial thickening positive case|Interlobular septal thickening positive case|
> |-|-|-|-|
> |16,695|530|518|381|
>
> This highlights that rare conditions have limited representation, and thus pruning or attention mechanisms may struggle in such areas.
>
> To address "statistical differences in token-dropping based on instructions", we provide quantitative evidence showing how instructions affect token retention in same organ. For lung-related tasks, the average number of retained tokens for Image Observation, Anomaly Detection, Medical Measurement, and Existence Detection is:
>
> ||Image Observation|Anomaly Detection|Medical Measurement|Existence Detection|
> |-|-|-|-|-|
> |Retained Tokens|485.3|479.6|499.5|455.9|
>
> This shows that different tasks lead to systematic adjustments in the retention ratio, with Measurement retaining more tokens and Existence Detection retaining fewer, indicating that pruning is conditioned on the task, not a fixed global rate.

---

> > ### Author Response · Authors · 2025-11-22
> > **Response to Reviewer buqX (3/3)**
> >
> > **W4.2: Failure case analysis and instruction-conditioned token pruning analysis.**
> >
> > For spatial distribution of retained tokens, we measure Jaccard dissimilarity (1 - overlap) for retained regions across different instructions. The dissimilarity between organ-related instructions is as follows:
> >
> > |kidney vs liver|liver vs spleen|kidney vs lung|
> > |-|-|-|
> > |0.6810|0.6646|0.5530|
> > |**liver vs lung**|**lung vs spleen**|**kidney vs spleen**|
> > |0.6852|0.5237|0.5247|
> >
> > The dissimilarity consistently falls between 0.52 and 0.69, demonstrating that different organ-related instructions lead to noticeable differences in the retention regions.
> >
> > We further validate that instruction-conditioned pruning preserves key regions by analyzing all lung-related instructions in 3D-RAD. We performed block-level overlap between retained tokens and reference lung mask segmented by pretrained models and resampled to our token grid. For each volume, we computed the fraction of retained lung-positive blocks and aggregated results across the full lung subset.
> >
> > The global and per-volume coverages show that the pruning module preserves most lung parenchyma with consistent behavior across cases. The resutls is reasonable since some lesion-specific cases do not require full-parenchyma observation.
> >
> > ||Global coverage|Mean per volume coverage|
> > |-|-|-|
> > |Ours 3B|0.831|0.818|
> >
> > All these results show that Photon’s token scheduling is indeed instruction-conditioned, and the current errors mainly stem from inherent difficulties in small-scale measurements and long-tail anatomical cases, rather than instability in the pruning mechanism.
> >
> > **Q4: Scaling to larger model.**
> >
> > Thank you for your suggestion. Due to time constraints, we have only extended our approach to a 7B model. As shown in the table below, the 7B model achieves a performance improvement over the 3B model, which confirms that our approach scales effectively with larger model sizes.
> >
> > ||Existence Detection|Static Temporal Diagnosis|Longitudinal Temporal Diagnosis|Medical Measurement|Image Observation|Anomaly Detection|Average Keep Token Num|
> > |-|-|-|-|-|-|-|-|
> > |Ours 3B|83.07|52.86|77.01|37.74/39.36/96.14|51.59/56.66/93.62|42.33/47.50/91.96|0.39K|
> > |Ours 7B|83.50|53.90|78.03|40.70/41.90/96.20|53.34/57.65/93.77|42.66/48.14/91.99|0.41K|
> >
> > **Q5: Evaluating on more archtectures.**
> >
> > Thank you for the suggestion. Beyond **Qwen2.5-VL 3B**, for the convenience of portability, we tested our method on **Qwen3-VL 2B**, which uses the newest Qwen3 backbone and a redesigned visual feature insert mechanism (Due to time constraints, this may not be the best practice, and is only for functional verification). We observed similar trends: our instruction-guided token scheduling reduces visual tokens without causing training collapse, suggesting that the method is not tied to a single Qwen2.5-VL architecture.
> >
> > |3D-RAD [1]|Existence Detection|Static Temporal Diagnosis|Longitudinal Temporal Diagnosis|Medical Measurement|Image Observation|Anomaly Detection|Average Keep Token Num|
> > |-|-|-|-|-|-|-|-|
> > |Photon (Qwen2.5-VL 3B)|83.07|52.86|77.01|37.74/39.36/96.14|51.59/56.66/93.62|42.33/47.50/91.96|0.39K|
> > |Photon  (Qwen3-VL 2B)|82.28|52.21|75.03|37.65/39.22/95.84|50.59/55.30/93.40|41.30/46.98/91.84|0.37K|
> >
> >
> > Applying the method to LLaMA-, Gemma- and InternLM-based MLLMs is a next step, and we plan to implement and evaluate these variants in our follow-up work.
> >
> >
> >
> > **Thank you for your valuable comment again, we hope our responses can address your concern and we look forward to your further reply!**
> >
> > Reference:
> >
> > [1] 3D-RAD, NeurIPS 2025, Gai et al.
> >
> > [2] NExT-QA, CVPR 2021, Xiao et al.

---

> > > ### Comment · Reviewer_buqX · 2025-11-25
> > >
> > > Thank the authors for their amazing and detailed reply. I am very glad to see these additional results and analysis. The new evaluation of the instruction-conditioned token pruning is very inspiring, and it serves as important evidence to support the claim of the original paper. The new visualization is also much better!
> > >
> > > The new analysis of the hyperparameter also somewhat reduced my concern about the complexity, but in some specific parameters like $\beta$, it can still lead to a difference greater than 2%, suggesting the method is still relatively sensitive to the parameter change. Yet, this does not harm the contribution of this paper; I believe it is still a wonderful work for the fields and is worth being presented to the community.
> > >
> > > Thus, I would decide to maintain my current score and recommend acceptance.

---

> > > > ### Author Response · Authors · 2025-11-26
> > > > **Thanks for recognizing the value of our work!**
> > > >
> > > > We sincerely thank the reviewer for recognizing the value of our contribution! Your assessment is very important to us and provides meaningful confirmation of our continued efforts to advance this area of research. We will keep refining the work and exploring broader applications in future studies. Thank you again for your thoughtful evaluation.

---

### Author Response · Authors · 2025-11-22
**General Response**

We extend our sincere gratitude to all the reviewers (R1-buqX, R2-goC1, R3-6q9R, and R4-TVyY) for their insightful and constructive reviews, which help us to emphasize the contributions of our approach. We are pleased to hear that the reviewers recognize the impressive performance improvements of our work (R1, R2, R3, R4), the well-structured presentation of our research (R4), the novelty of our proposed methods (R1, R2, R3), as well as the sufficient transparent ablation experimental verification (R1, R2, R3, R4).

**In direct response to your thoughtful comments, we have methodically addressed each point in our individual responses, and we provide a summary here:**

+ We corrected writing issues, refined figures and notations, and added higher-level explanations of ITS, SGP, and their hyperparameters to improve clarity and reproducibility.
+ We added direct comparisons with recent token pruning baselines and new results on other backbones to support the claimed efficiency and scalability of Photon.
+ We conducted new experiments on additional datasets and modalities to validate the robustness and generalization ability of our method across different tasks and data distributions.
+ We expanded robustness, statistical, and anatomical analyses, including noisy-threshold studies, multi-seed and bootstrap evaluation, and overlap with organ segmentations, to show that Photon yields stable gains and preserves clinically important regions under dynamic pruning.

**Thanks again for all of your valuable suggestions,  we have revised the paper accordingly, updating the main text, figures, and appendix to reflect the above changes.**

We appreciate the reviewers' time to check our response and hope to further discuss with the reviewers whether the concerns have been addressed or not. If the reviewers still have any unclear parts about our work, please let us know.



Best Regards,

All #3675 Authors

---

### Meta-Review · Area_Chair_sFBV · 2026-01-07

**Summary:**

The reviewers generally recognize the novelty and effectiveness of the proposed method, Photon. The rebuttal has satisfactorily addressed concerns regarding comparisons with additional pruning baselines, hyperparameter sensitivity, and the clarity of writing and presentation. The revised manuscript demonstrates improved clarity and readability.

However, the revisions were not highlighted, making it difficult to identify the specific changes made. Additionally, the authors should fulfill their commitment to release the entire codebase to ensure full reproducibility and to support future research in the field.

**Reviewer Concerns:**

In their rebuttal and revised manuscript, the authors presented new results and analyses on instruction-conditioned token pruning, included comparisons with additional pruning baselines, and provided detailed hyperparameter settings to support reproducibility. They also improved the visualizations, making them clearer and more effective. The new hyperparameter analysis addresses previous concerns about the method’s complexity and reproducibility.

I do not see any significant remaining issues. However, the authors should ensure they release the complete codebase as promised, to guarantee full reproducibility and support future research.

**Reviewer Scores:**

- Reviewer buqX, after reading the author response, confirmed they will maintain their positive rating of 8.
- Reviewer goC1 is likely to increase their initial rating from 4 to 6, as they found the author response detailed and convincing.
- Reviewer 6q9R initially gave a positive rating of 6 and indicated a willingness to raise their score if the rebuttal addressed their concerns; given the quality of the rebuttal, 6q9R is likely to either raise or maintain their positive rating.
- Reviewer TVyY gave an initial rating of 4, but may reconsider if they are satisfied with the rebuttal’s explanations regarding novelty and reproducibility.

Overall, it appears that the paper is likely to receive three positive ratings after the rebuttal.

---

### Decision · Program_Chairs · 2026-01-26

Accept (Poster)